# *Pten* is necessary for the quiescence and maintenance of adult muscle stem cells

Feng Yue[1], Pengpeng Bi[1], Chao Wang[1], Tizhong Shan[1], Yaohui Nie[1,2], Timothy L. Ratliff[3,4], Timothy P. Gavin[2] & Shihuan Kuang[1,3]

Satellite cells (SCs) are myogenic stem cells required for regeneration of adult skeletal muscles. A proper balance among quiescence, activation and differentiation is essential for long-term maintenance of SCs and their regenerative function. Here we show a function of Pten (phosphatase and tensin homologue) in quiescent SCs. Deletion of *Pten* in quiescent SCs leads to their spontaneous activation and premature differentiation without proliferation, resulting in depletion of SC pool and regenerative failure. However, prior to depletion, *Pten*-null activated SCs can transiently proliferate upon injury and regenerate injured muscles, but continually decline during regeneration, suggesting an inability to return to quiescence. Mechanistically, *Pten* deletion increases Akt phosphorylation, which induces cytoplasmic translocation of FoxO1 and suppression of Notch signalling. Accordingly, constitutive activation of Notch1 prevents SC depletion despite *Pten* deletion. Our findings delineate a critical function of Pten in maintaining SC quiescence and reveal an interaction between Pten and Notch signalling.

[1] Department of Animal Sciences, Purdue University, 901 W State Street, West Lafayette, Indiana 47907, USA. [2] Department of Health and Kinesiology, Purdue University, 800 W. Stadium Ave, West Lafayette, Indiana 47907, USA. [3] Center for Cancer Research, Purdue University, 201 S University Street, West Lafayette, Indiana 47907, USA. [4] Department of Comparative Pathobiology, Purdue University, 625 Harrison St, West Lafayette, Indiana 47907, USA. Correspondence and requests for materials should be addressed to S.K. (email: skuang@purdue.edu).

Tissue-specific adult stem cells are capable of regenerating local tissues continuously throughout life. Defining features of stem cells include the ability to differentiate into mature cell types and to retain stem cell identity by self-renewal[1]. Adult skeletal muscles have a robust regenerative capacity, relying on a population of resident stem cells called satellite cells (SCs)[2,3]. SCs are mitotically quiescent in adult health skeletal muscles and reside in a sublaminar niche adjacent to the host myofiber. Quiescent SCs (QSCs) can be identified by the unique expression of Pax7 in the muscle[4], and thus several lines of $Pax7^{Cre}$ or $Pax7^{CreER}$ mice have been commonly used to label SCs and their descendants[5]. In response to injury or growth factor stimulation, SCs are activated and proliferate extensively[6,7]. Following proliferation, a majority of SC progeny undergo myogenic terminal differentiation and fuse together for de novo myotube formation, or fuse with damaged myofibers to repair the injury[7,8]. Meanwhile, a subset of proliferating SCs withdraws from the cell cycle and returns to the quiescent state to maintain the stem cell pool[7,8]. The self-renewing, proliferating and differentiating SC progenies can be reliably identified as $Pax7^+/MyoD^-$, $Pax7^+/MyoD^+$ and $Pax7^-/MyoD^+$, respectively[9–11]. The fate choices of SCs have been found to be regulated by a number of signalling molecules, including Notch[12–14], Wnt[15,16], Lkb1 (ref. 17), sirtuin 1 (ref. 18), cytokines[19] and non-coding RNAs (miR-489)[20] among others[21–24]. However, mechanisms governing the quiescent state of SCs are poorly understood.

The phosphatase and tensin homologue (Pten), encoding a dual-specificity lipid and protein phosphatase, was originally identified as a tumour suppressor gene mutated in a wide range of malignancies[25,26]. The canonical function of Pten is to antagonize PI3K and dephosphorylate Akt, therefore inhibiting downstream mTOR signalling that positively regulates cell growth and survival[27]. Pten has important roles in regulating functions of neural and haematopoietic stem cells (HSCs), including their self-renewal and differentiation[28–31]. Conditional deletion of Pten in adult neural stem cells leads to persistently enhanced self-renewal without signs of exhaustion[29]. However, conditional deletion of Pten in adult HSCs causes short-term expansion but long-term exhaustion of HSCs, resulting in the development of myeloproliferative disorder and leukaemia[30,32]. The known pleiotropic effects of Pten on various cell types suggest it may have essential but distinct cell context-dependent roles in different types of stem cells.

In skeletal muscles, Pten knockout (KO) in mature skeletal muscles driven by MCK-Cre does not lead to any obvious histological abnormality[33,34]; however, myogenic progenitor-specific Myf5-Cre driven Pten KO fails to delete Pten in limb muscles[35]. Therefore, the role of Pten in muscle stem cells and progenitor cells remains unknown. Here, we use the tamoxifen (TMX)-inducible $Pax7^{CreER}$ knockin allele to specifically delete Pten in QSCs in adult mice. Pten-null SCs spontaneously exit quiescence and undergo terminal differentiation without proliferation, leading to rapid exhaustion of the SC pool and failure of muscle regeneration. We find that depletion of Pten-null SCs is independent of mTOR activation, but is owing to inhibition of Notch signalling. Accordingly, activation of Notch signalling rescues the depletion of Pten-null SCs. These results indicate that Pten is required for maintaining the quiescence of adult muscle stem cells.

## Results

### Pten is expressed abundantly in quiescent and activated SCs.
As the first step to understand how Pten functions in SCs and their progeny, we examined Pten expression in quiescent, activated, proliferating and differentiating SCs attached on single myofibers isolated from the extensor digitorum longus (EDL) muscles of adult mice. Pten immunofluorescence was readily detectable in $Pax7^+$ QSCs located on freshly isolated EDL myofibers that were immediately fixed after isolation (Day 0 in Fig. 1a). We subsequently cultured the EDL myofibers in suspension for 1–3 days, during which SCs activate (Day 1), proliferate and differentiate (Day 2–3). Pten signal intensity remained high in activated SCs (Day 1), and then was more diffusely expressed in clusters of SCs progenies (Day 2–3), with higher signal intensity colocalized to $Pax7^+$ signal (Fig. 1a).

Next, we isolated SC-derived primary myoblasts from adult mice and determined Pten expression during their proliferation and differentiation. Pten was ubiquitously expressed in proliferating primary myoblasts cultured in growth medium (Fig. 1b). Upon induction of differentiation by serum withdrawal, however, Pten expression declined rapidly within 24 h and was undetectable within 72 h (Fig. 1b). Notably, downregulation of Pten corresponded to concomitant upregulation of MyoG and myosin heavy chain (marked by MF20), markers of myogenic differentiation (Fig. 1b). Consistent with the immunocytochemistry labelling, western blotting confirmed the concomitant downregulation of Pten, Pax7 and MyoD, and up regulation of MyoG and MF20 during myoblast transition from proliferation to differentiation (Fig. 1c). These data indicate that Pten expression is high in quiescent and activated SCs but low in differentiated myotubes.

### Loss of Pten leads to depletion of quiescent SC pool.
The identification of abundant Pten expression in adult SCs prompted us to explore its potential function in these cells. To achieve this goal, we generated SC-specific Pten KO mice by crossing $Pax7^{CreER}$ mice with $Pten^{f/f}$ mice in which exon 5 encoding the phosphatase domain of Pten was flanked by engineered LoxP sites. Genetic inactivation of Pten was induced by repeated intraperitoneal (IP) injection of TMX in adult $Pax7^{CreER}::Pten^{f/f}$ mice ($Pten^{PKO}$), using TMX-treated $Pten^{f/f}$ littermates as wild-type (WT) control. Co-immunostaining of Pax7 and Pten indicated that five consecutive daily injections of TMX followed by 7 days of chasing effectively eliminated Pten protein in $Pten^{PKO}$ but not WT SCs (Supplementary Fig. 1a). Overall, only <3% of SCs in $Pten^{PKO}$ mice still had Pten expression, whereas >99% of SCs expressed high levels of Pten in WT mice (Supplementary Fig. 1a). These results confirm the efficiency of our Pten conditional KO mouse model.

Using this model, we first examined how Pten KO affects QSCs in adult resting skeletal muscles. After 5 consecutive daily TMX injection followed by 4–28 days of chasing, we found a surprisingly rapid decline of QSCs in freshly isolated EDL myofibers of $Pten^{PKO}$ mice, but not WT mice (Fig. 2a,b). Specifically, ~50% of QSCs were lost within 7 days, ~80% were lost within 12 days and >90% were lost within 28 days in the $Pten^{PKO}$ mice (Fig. 2b). Similarly, immunofluorescence staining of Pax7 and α-laminin in tibialis anterior (TA) muscle cross-sections indicated robust ablation (~80%) of $Pax7^+$ QSCs in $Pten^{PKO}$ mice at Day 21 after TMX induction (Fig. 2c). These results demonstrate that loss of Pten leads to depletion of QSCs in adult resting muscle.

### Pten deletion in QSCs impairs muscle regeneration.
To further explore the functional significance of Pten in SCs, we examined the regenerative capacity of Pten KO SCs. TA muscles of $Pten^{PKO}$ and WT mice were induced to regenerate by cardiotoxin (CTX) following TMX-induction of Pten deletion. TA muscles of

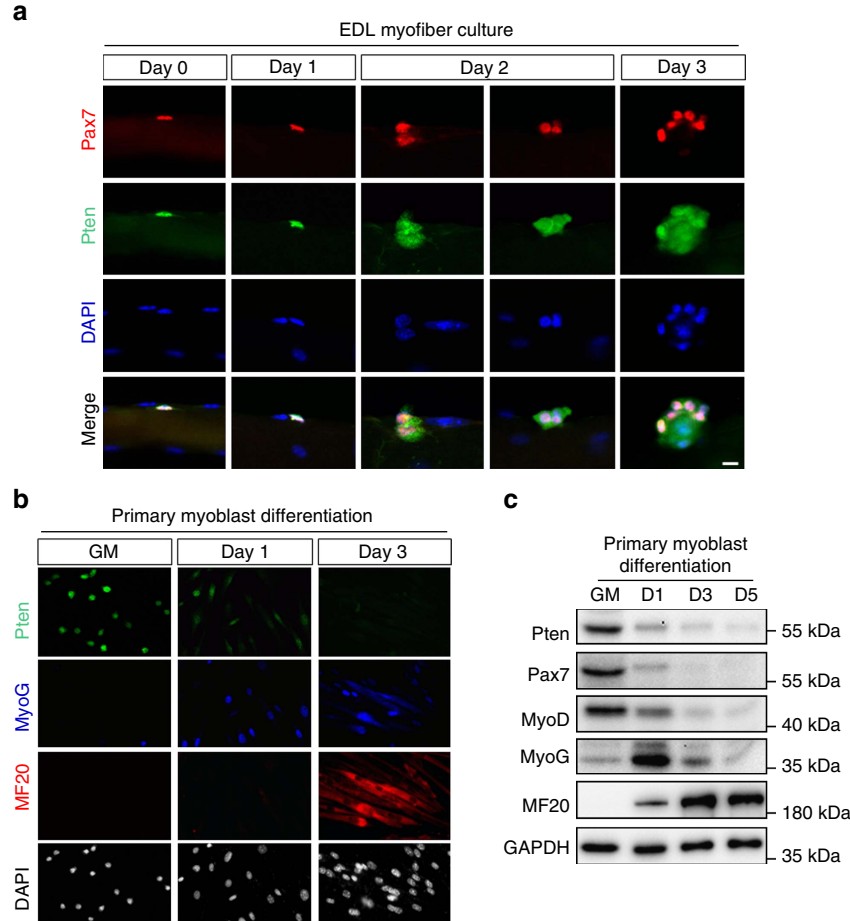

**Figure 1 | *Pten* is expressed abundantly in quiescent and activated SCs. (a)** Pten immunofluorescence in Pax7[+] SCs attached on freshly isolated EDL myofibers (Day 0) or after cultured for 1–3 days. Scale bar, 10 μm. **(b)** Co-immunostaining of Pten, MyoG (differentiation marker) and MF20 (myosin heavy chain) in primary myoblasts in growth medium (GM) or differentiated for 1–3 days. Scale bar, 50 μm. **(c)** Western blot showing relative levels of Pten and myogenic marker proteins at various stages of myogenic differentiation.

*Pten*$^{PKO}$ and WT mice regenerated equally well when CTX was injected at Day 4 after TMX induction, manifested by similar muscle weight and histology (Supplementary Fig. 1b–d). The relative normal regeneration is probably due to low percentage of SC ablation at 4 days after TMX induction (Fig. 2b). However, the number of Pax7[+] cells was reduced by ∼74% in *Pten*$^{PKO}$ compared with WT mice at the completion of muscle regeneration, similar to the extent of SC depletion (∼78%) in the non-injured muscles (Supplementary Fig. 1e). This observation suggests that *Pten*-null SCs undergo continuous depletion during regeneration, and fail to return to quiescence and maintain the SC pool.

When TA muscles were injured at Day 21 after TMX induction (at which time point >80% SCs were depleted, Fig. 2c), *Pten*$^{PKO}$ mice regeneration was extremely poor (Fig. 2d,e). Compared with regenerated muscles in WT mice, *Pten*$^{PKO}$ mice had reduced muscle weight, reduced number of regenerated myofibers and myofiber size, smaller regenerated area and lower levels of dystrophin expression, but increased fibrosis and mIgG infiltration indicative of inflammation (Fig. 2f–j and Supplementary Fig. 2a). Strikingly, following acute injury the number of SCs in *Pten*$^{PKO}$ relative to WT mice was reduced by 88% (Supplementary Fig. 2b), indicating that the remnant SCs in *Pten*$^{PKO}$ mice cannot repopulate the SC pool even in response to proliferative cues during regeneration. These data suggest that although *Pten*-null SCs (prior to depletion) can repair muscle injury, they are incapable of returning to quiescence and maintaining SC pool.

Given that 2–3% of SCs escaped *Pten* deletion at Day 7 after TMX induction, we asked if these remnant SCs could restore the SC pool in the long term. To answer this, we first examined the number of SC in uninjured TA muscles 4 months after TMX induction (Supplementary Fig. 3a). Notably, the number of Pax7[+] SCs remained at ∼3% in relative WT mice (0.06 versus 1.92 cells per cross-sectional area, Supplementary Fig. 3b). We then examined if the remnant SCs could repopulate during regeneration. As expected, the muscles of *Pten*$^{PKO}$ mice regenerated extremely poorly 14 days after two consecutive CTX injuries, revealed by reduced muscle size and the number of regenerated myofiber (Supplementary Fig. 3c,d). Moreover, the number of Pax7[+] SCs in injured TA muscle of *Pten*$^{PKO}$ mice was still ∼2% of that in WT mice (0.08 versus 3.62 cells per TA area, Supplementary Fig. 3e). These results suggest that the remnant Pten[+] SCs fail to repopulate SC pool and repair injured muscles even after long-term recovery.

**Pten-null QSCs lose quiescence and undergo differentiation.** We next questioned whether *Pten* KO leads to loss of SCs through cell death or cell differentiation. To test this, we labelled apoptotic cells by *in situ* terminal deoxynucleotidyl transferase dUTP nick-end labelling (TUNEL). As positive control, TUNEL[+]

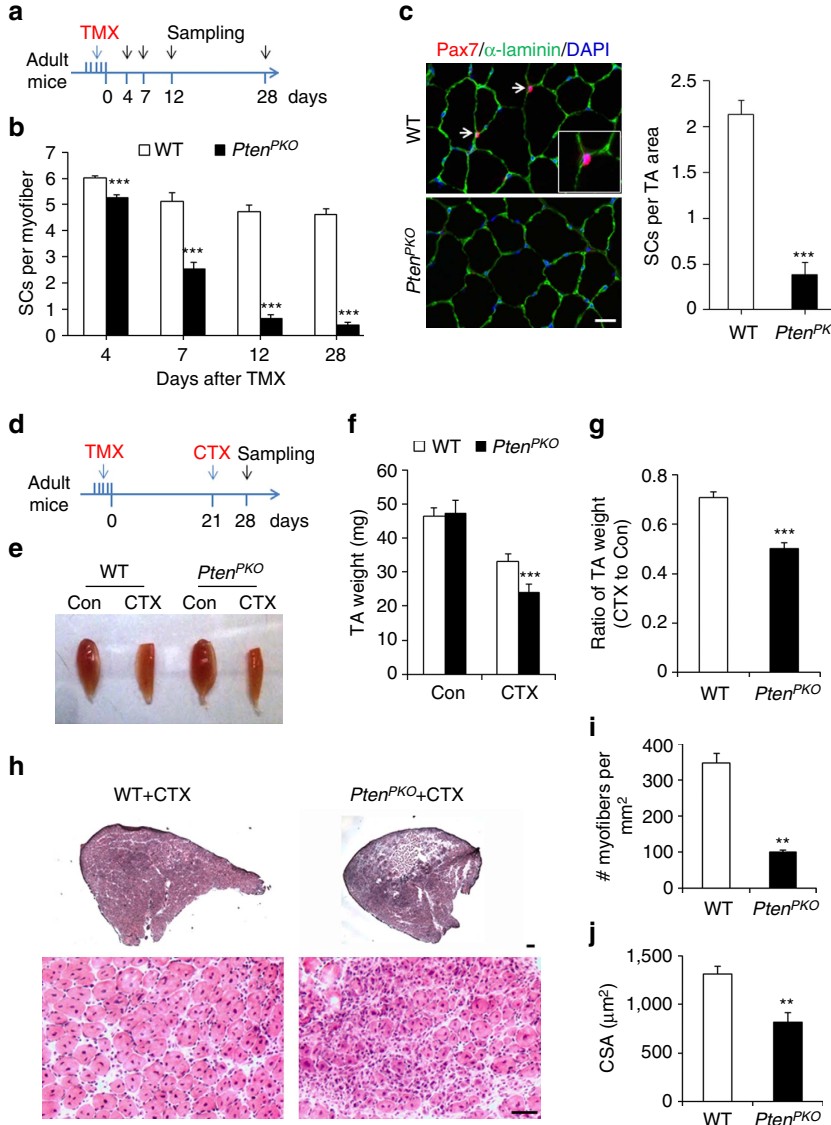

**Figure 2 | *Pten* deletion in QSCs depletes SC pool and impairs muscle regeneration.** (**a**) Schematics showing timing of tamoxifen (TMX) induction and sample collection. (**b**) Number of SCs per fresh EDL myofiber of WT and *Pten*[PKO] mice 4–28 days after TMX induction. Data represent mean ± s.e.m. (*t*-test: ***P < 0.001; Day4, n = 4; Day7, n = 6; Day12, n = 4; Day28, n = 5, each group; 21 myofibers per animal). (**c**) Left panel, immunofluorescence of Pax7 and α-laminin in TA muscle cross-sections 21 days after TMX induction. White arrow indicates SCs. Scale bar: 20 μm. Right panel, the average number of SCs per microscopic area. Data represent mean ± s.e.m. (*t*-test: ***P < 0.001; n = 5, each group). (**d**) Schematics showing TMX induction and CTX injection. (**e**) Representative images of TA muscles in WT and *Pten*[PKO] mice. (**f,g**) TA muscle weight (**f**) and ratio of injured muscle weight (**g**). Data represent mean ± s.e.m. (*t*-test: ***P < 0.001; n = 7, each group). (**h**) H&E staining of TA muscle cross-sections (upper panels, scale bar: 200 μm) and magnified representative regenerated areas (bottom panels, scale bar: 50 μm). (**i,j**) Number of regenerated myofibers per mm² (**i**) and average cross-sectional area (CSA) of regenerated myofibers (**j**) in TA muscle cross-sections. Data represent mean ± s.e.m. (*t*-test: **P < 0.01; n = 4, each group).

SCs were observed on fresh EDL myofibers treated with DNaseI (Supplementary Fig. 4a). However, TUNEL+ SCs were not detected on myofibers of WT (n = 522 SCs) or *Pten*[PKO] (n = 521 SCs) mice 7 days after TMX induction (Fig. 3a). Similar results were observed on TA muscle cross-sections (Supplementary Fig. 4b). Consistently, <2% of SCs on myofibers was positively labelled by cleaved-caspase3, an early marker of apoptotic cells, in WT or *Pten*[PKO] mice (Fig. 3b). These results demonstrate that *Pten* KO-induced SC depletion was not due to cell apoptosis.

We further asked if *Pten* KO leads to premature differentiation, thus depleting SCs. To test this possibility, we examined the expression of SC activation marker MyoD and differentiation marker MyoG at Day 2, 4 and 9 after TMX induced deletion of *Pten* in the *Pten*[PKO] mice. Indeed, we found emergence of MyoD+, MyoG+ and MyoD+/MyoG+ (double positive) cells on fresh isolated EDL myofibers from the *Pten*[PKO] mice (Fig. 3c), but not WT mice (Supplementary Fig. 5a). Similar results were observed on TA muscle cross-sections (Supplementary Fig. 5b). Analysis of the time course of MyoD and MyoG expression indicated that abundance of MyoD+ cells peaked at Day 4, and abundance of MyoG+ cells peaked at Day 9 after TMX induction (Fig. 3d,e). Consistent with the immunostaining results, higher mRNA levels of *MyoD* and *MyoG* were detected in *Pten*[PKO] than WT muscles at Day 9 after TMX induction (Supplementary Fig. 5c). Thus, *Pten* deletion

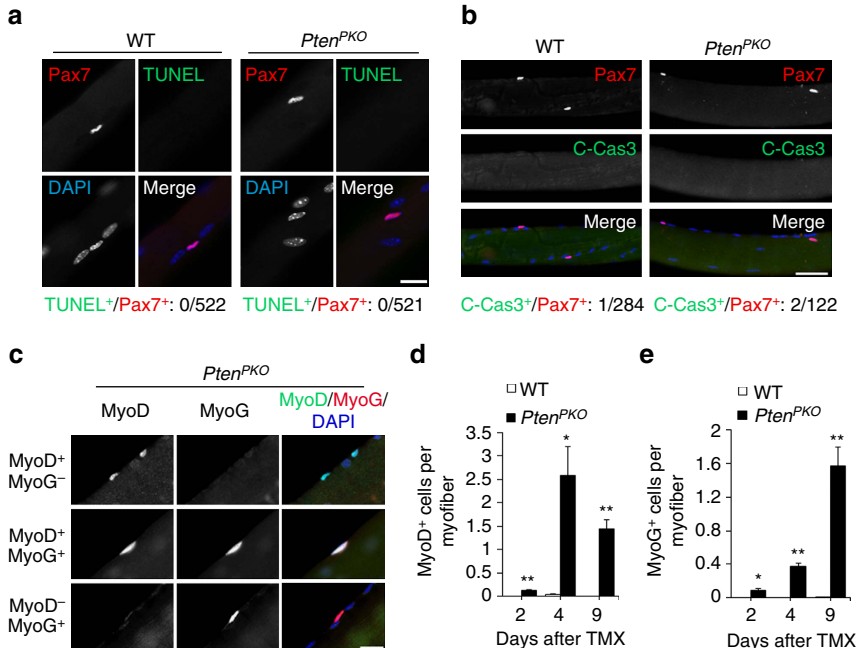

**Figure 3 | Depletion of *Pten*-null SCs is not due to cell apoptosis but due to differentiation.** (**a**) Immunofluorescence of TUNEL in SCs on fresh EDL myofibers isolated from WT and *Pten*^PKO mice 7 days after TMX induction. Scale bar, 20 μm. Quantification analysis of total 522 SCs in WT and 521 SCs in *Pten*^PKO mice (n = 3 animals, each group). (**b**) Immunofluorescence of Cleaved-Caspase3 (Green) and Pax7 (Red) on fresh EDL myofibers. Scale bar, 50 μm. Quantification analysis of total 284 SCs in WT and 122 SCs in *Pten*^PKO mice (n = 3 animals, each group). (**c**) Immunofluorescence of MyoD and MyoG on freshly isolated EDL myofibers of *Pten*^PKO mice 4 or 9 days after TMX induction. Scale bar: 20 μm. (**d,e**) Quantification of the average numbers of MyoD^+ (**d**) and MyoG^+ (**e**) cells per EDL myofiber in WT and *Pten*^PKO mice at the time shown. Data represent mean ± s.e.m. (t-test: *P < 0.05; **P < 0.01; Day2, n = 4; Day4, n = 3; Day9, n = 4, each group; 20 myofibers per animal).

leads to spontaneous activation and premature differentiation of QSCs in adult resting muscle.

**Pten-null QSCs differentiate without proliferation.** To examine if the activated *Pten* KO SCs have undergone cell cycle entry and proliferation before differentiation, we performed 5-ethynyl-2′-deoxyuridine (EdU) incorporation assay. Mice received uninterrupted EdU administration through drink water beginning at the onset of TMX induction until being analysed (Supplementary Fig. 6a). As a positive control, EdU marked many cells in the highly proliferative intestinal epithelia (Supplementary Fig. 6b), thus confirming the effectiveness of EdU labelling. Despite fewer numbers of total SCs at 7 days after TMX induction (Fig. 2b), a significantly higher number of EdU^+ SCs were found on fresh EDL myofibers of *Pten*^PKO compared with WT mice (0.82 versus 0.37 per myofiber in *Pten*^PKO and WT, respectively), this corresponds to 3.2 times increase in percentage (20.2% versus 6.3% in *Pten*^PKO and WT, respectively) (Fig. 4a–c). These results suggest that a subset of *Pten*-null SCs have undergone S-phase entry, but have not gone through the cell division as two sister EdU^+ SCs were never detected on the same myofiber. To further confirm this, we labelled the SCs with the mitotic marker Ki67 and metaphase marker phospho-Histone H3 (pHH3). The abundance of Ki67^+ SCs was indistinguishable in WT and *Pten*^PKO muscles 7 days after TMX induction (Supplementary Fig. 7a,b). Consistently, pHH3^+ SCs were not detected in WT (n = 444 SCs) or *Pten*^PKO muscles (n = 392 SCs, Supplementary Fig. 7c). These results indicate that loss of *Pten* leads to increased cell cycle entry of QSCs without completing cell division.

We further examined the fate of the EdU-labelled SCs. Co-labelling of MyoG and EdU revealed a significant increase in MyoG^+ EdU^+ cells in *Pten*^PKO mice (0.91 per myofiber) compared with WT mice (0.07 per myofiber) 7 days after TMX induction (Fig. 4d,e). Notably, 71% and 92% of the MyoG^+ cells were EdU^+ in WT and *Pten*^PKO mice, respectively (Fig. 4f), suggesting that the EdU-labelled SCs have differentiated. To further test if the differentiated MyoG^+ cells fused into myofiber, we labelled the myofiber membrane with dystrophin in TA muscle cross-sections. Some EdU^+ nuclei were detected under the myofiber membrane in both WT and *Pten*^PKO mice (Fig. 4g and Supplementary Fig. 8a), indicating that they have fused into myofibers and became myonuclei. Strikingly, the number of EdU^+ myonuclei was significantly higher in *Pten*^PKO (∼8.7 per TA area) compared with WT mice (∼1.3 per TA area) 7 days after TMX induction (Fig. 4h), suggesting accelerated differentiation and fusion of *Pten*-null SCs. Moreover, the total myonuclei number in EDL myofibers was comparable between WT and *Pten*^PKO mice 18 days after TMX induction (Supplementary Fig. 8b), arguing against the possibility that extensive proliferation occurred prior to fusion of in *Pten*-null SCs. In contrast to the central location of myonuclei in regenerating myofibers, the EdU^+ myonuclei were located peripherally in uninjured myofibers (Fig. 4g and Supplementary Fig. 8a), and the abundance of centronuclei was indistinguishable in WT and *Pten*^PKO mice 7 days after TMX induction (Supplementary Fig. 8c). Taken together, these results demonstrate that *Pten*-null SCs lose quiescence, then differentiate and fuse into myofibers without proliferation.

**Activation of Akt and mTOR pathways in *Pten*-deficient SCs.** We next investigated the molecular mechanism mediating *Pten* KO-induced SC activation and ablation. Previous studies indicate

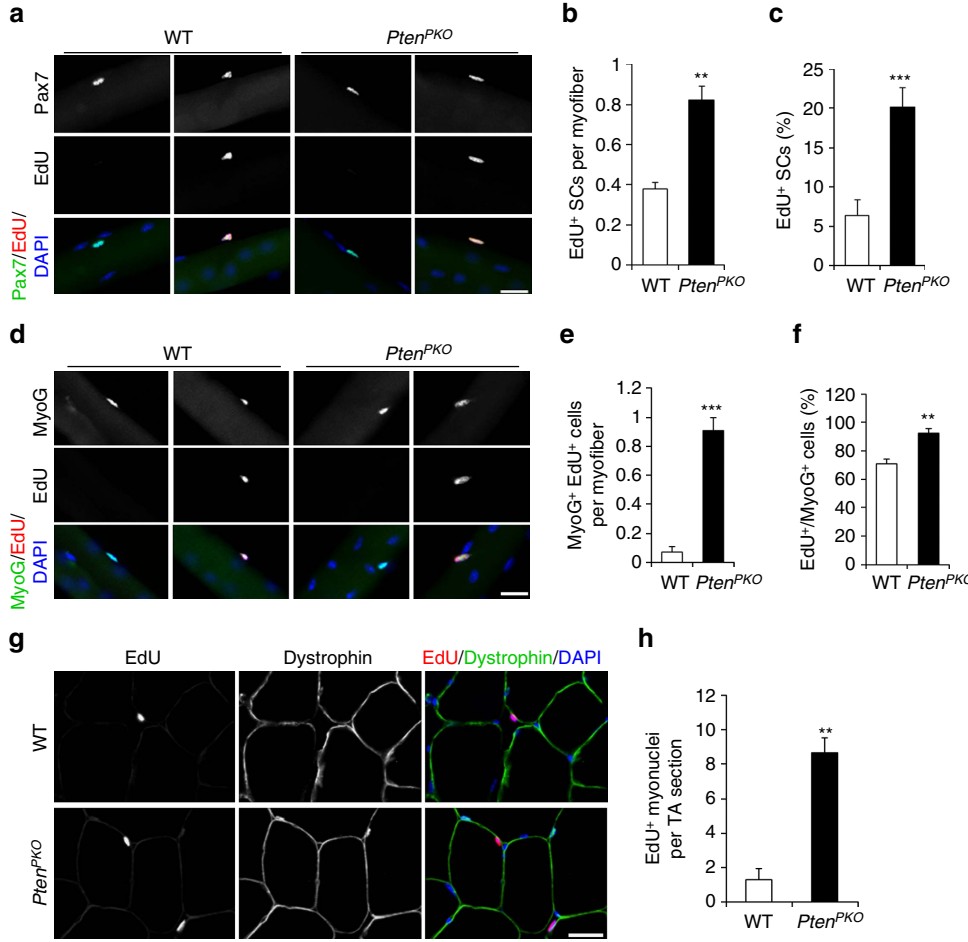

**Figure 4 | *Pten*-null QSCs initiate terminal differentiation with S-phase entry but bypassing proliferation.** (**a**) Immunostaining of Pax7 and EdU on freshly isolated EDL myofibers of WT and *Pten*^PKO^ mice 7 days after TMX induction and EdU incorporation. Scale bar: 20 μm. (**b,c**) Quantification of the number of EdU $^+$ SCs per EDL myofiber (**b**) and the percentage of EdU $^+$ SCs (**c**). Data represent mean ± s.e.m. (*t*-test: **$P < 0.01$; ***$P < 0.001$; $n = 6$, each group; 40 myofibers per animal). (**d**) Immunostaining of MyoG and EdU on freshly isolated EDL myofibers. Scale bar: 20 μm. Quantification of the number of MyoG $^+$ EdU $^+$ cells per EDL myofiber (**e**) and the percentage of MyoG $^+$ EdU $^+$ cells in MyoG $^+$ cells (**f**). Data represent mean ± s.e.m. (*t*-test: **$P < 0.01$; ***$P < 0.001$; $n = 6$, each group; 27 myofibers per animal). (**g**) Immunostaining of EdU and dystrophin on TA muscle cross-sections 7 days after TMX induction and EdU incorporation. Muscle was fixed before freeze and cryosection. Scale bar: 20 μm. (**h**) Quantification of the EdU $^+$ myonuclei number on TA muscle cross-sections. Data represent mean ± s.e.m. (*t*-test: **$P < 0.01$; $n = 6$, each group).

that activations of hematopoietic and hair follicle stem cells from quiescence are dependent on the Akt-mTOR signalling[36–39]. We therefore examined the phosphorylation of Akt (pAkt, Fig. 5a), a key step in the activation of Akt. At Day 4 after TMX induction, Pax7$^+$ SCs on WT myofibers were mostly (∼80%) pAkt negative (pAkt$^{Neg}$) and the remaining 20% expressed low levels of pAkt (pAkt$^{Low}$). By contrast, most *Pten*$^{PKO}$ SCs (∼60%) expressed high levels of pAkt (pAkt$^{High}$) and only ∼10% of Pten KO SCs were pAkt$^{Neg}$ (Fig. 5b). We also examined the phosphorylation of S6 (pS6) (Fig. 5c,d), a direct target of Akt-mTOR pathway. Whereas the WT myofibers contained roughly equal numbers of pS6$^{Neg}$ and pS6$^{Low}$ SCs, the *Pten*$^{PKO}$ myofibers carried predominantly pS6$^{High}$ SCs (Fig. 5c,d). These results demonstrate that *Pten* KO leads to the activation of Akt-mTOR signalling in SCs.

**Inhibition of mTOR does not rescue depletion of *Pten*-null SCs.** As *Pten* KO activates Akt-mTOR signalling, we examined whether inhibition of mTOR signalling could rescue SC depletion and regenerative defects of the *Pten*$^{PKO}$ mice. We IP injected

adult WT and *Pten*$^{PKO}$ mice with rapamycin, a highly specific pharmacological inhibitor of mTOR, starting from the first day of TMX induction and continuously throughout the chasing and CTX-induced muscle regeneration period (Fig. 6a). One day prior to harvesting muscle, the mice were IP injected with Evans blue dye to visualize the damaged muscle fibres. After injury, much more Evans blue accumulation was observed in TA muscles of *Pten*$^{PKO}$ mice compared with WT regardless of rapamycin treatment, indicating more damaged myofibers in *Pten*$^{PKO}$ muscles (Fig. 6b). Consistently, immunostaining by mIgG and dystrophin showed that the TA muscles of *Pten*$^{PKO}$ mice contained fewer dystrophin positive myofibers and more severe fibrosis than did the WT, again regardless of rapamycin treatment (Fig. 6c). Overall, rapamycin administration failed to rescue the regenerative defects of the *Pten*$^{PKO}$ mice.

We further examined whether rapamycin could rescue SC depletion after *Pten* KO. At Day 21 after TMX induction, Pax7$^+$ SCs were almost completely eliminated in the *Pten*$^{PKO}$ mice, and rapamycin not only failed to rescue SC depletion in the *Pten*$^{PKO}$ mice but also reduced SCs in WT mice (Fig. 6d). Specifically, in the absence of rapamycin, *Pten* KO reduced

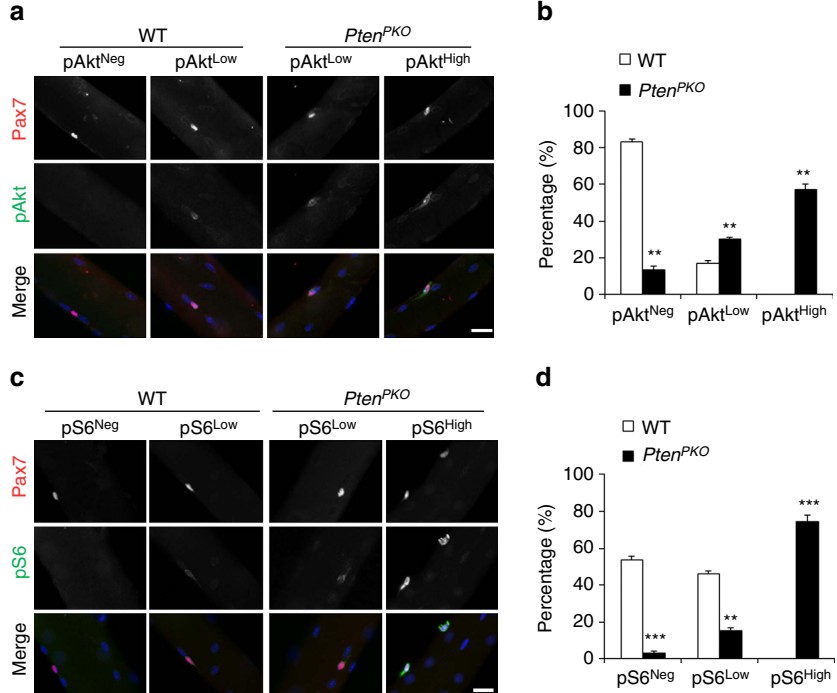

**Figure 5 | Activation of Akt and mTOR pathways in *Pten*-deficient SCs.** (**a**) Immunofluorescence of pAkt in SCs on freshly isolated EDL myofibers of WT and *Pten*$^{PKO}$ mice 4 days after TMX induction. Scale bar, 20 μm. (**b**) Quantification of the percentages of pAkt$^{Neg}$, pAkt$^{Low}$ and pAkt$^{High}$ SCs shown in (**a**). Data represent mean ± s.e.m. (*t*-test: \*\**P* < 0.01; *n* = 3, each group; 15 myofibers per animal). (**c**) Immunofluorescence of pS6 in SCs on freshly isolated EDL myofibers of WT and *Pten*$^{PKO}$ mice 4 days after TMX induction. Scale bar, 20 μm. (**d**) Quantification of the percentages of pS6$^{Neg}$, pS6$^{Low}$ and pS6$^{High}$ SCs shown in (**c**). Data represent mean ± s.e.m. (*t*-test: \*\**P* < 0.01; \*\*\**P* < 0.001; *n* = 3, each group; 15 myofibers per animal).

the number of SCs by 88%, whereas in the presence of rapamycin, *Pten* KO reduced the number of SCs by 93% (Fig. 6d). Likewise, after CTX-induced muscle regeneration, the *Pten*$^{PKO}$ muscles had similarly low numbers of SCs with or without rapamycin treatment (Fig. 6e). Together, these results indicate that depletion of adult SC pool induced by *Pten* KO is independent of mTOR signalling.

***Pten* KO inhibits Notch signalling through Akt-FoxO1**. Besides mTOR, another main substrate of pAkt is Forkhead box protein O1 (FoxO1), whose phosphorylation suppresses its transcriptional activity through cytoplasmic translocation[40]. We therefore asked if increased pAkt would change the intracellular localisation of FoxO1 in the *Pten* KO SCs. In controls, FoxO1 was predominantly located in the nucleus of SCs (Fig. 7a,b). At Day 4 after TMX-induced *Pten* KO in SCs, however, FoxO1 signal was detected in the cytoplasm of Pax7$^+$ SCs (Fig. 7a,b). The nuclear to cytoplasmic translocation of FoxO1 is due to *Pten* deletion in Pax7$^+$ SCs as FoxO1 signal remained in the nucleus of Pax7$^-$ myonuclei in which *Pten* should not have been deleted (Fig. 7a,b). Overall, FoxO1 was localized in the nucleus of ~98% WT SCs. By contrast, ~93% *Pten*$^{PKO}$ SCs have FoxO1 localized in the cytoplasm and the remaining 7% in the nucleus (Fig. 7c). The nuclear to cytoplasmic translocation of FoxO1 is accompanied by increased levels of pFoxO1 and pAkt (Fig. 7c). These results suggest that activation of Akt in *Pten* KO SCs phosphorylates FoxO1 and leads to its cytoplasmic translocation.

To understand how FoxO1 intracellular localisation regulates SC dynamics, we focused on Notch signalling, a key governor of SC quiescence[13,14]. Because Akt activation has been shown to inhibit Notch signalling through suppression of FoxO[41–43],

we determined whether *Pten* KO or overexpression affected Notch signalling. We isolated primary myoblasts from *Pax7Cre*$^{ER}$::*Pten*$^{f/f}$ mice and used 4-OH-TMX to induce *Pten* deletion in the myoblasts. Interestingly, expression of most Notch target and ligand genes was downregulated in *Pten* KO myoblasts compared with control myoblasts treated with vehicle (Fig. 7d). Consistent with gene expression at the mRNA level, western blotting also demonstrated a dramatic decrease of Hes1 but not Hey1 protein in *Pten* KO myoblasts (Fig. 7e). Conversely, overexpression of Pten in cultured primary myoblasts upregulated the expression of several Notch downstream genes (Supplementary Fig. 9a).

We next investigated the involvement of FoxO1 in mediating Notch signalling downstream of Pten. Indeed, overexpression of the constitutively active FoxO1 (FoxO1-ADA) in primary myoblasts upregulated the expression of several Notch target genes (Supplementary Fig. 9b,c), suggesting that FoxO1 positively regulates Notch signalling. Previously, FoxO1 was reported to regulate Notch signalling through interacting with RBPJκ, a nuclear mediator of Notch signalling[43]. We confirmed the endogenous interaction between FoxO1 and RBPJκ in primary myoblasts (Supplementary Fig. 9d), and the interaction was significantly enhanced by activation of Notch signalling or overexpression of FoxO1-ADA (Fig. 7f). To examine if the interaction of FoxO1-RBPJκ affects expression of Notch target genes, we measured RBPJκ-induced luciferase activity in the presence or absence of FoxO1-ADA in HEK293A cells. Transfection of a constitutively active form of RBPJκ by fusion with VP16 transactivation domain (pRBPJκ-VP16) markedly increased the 4xCSL-luciferase activity compared with the control transfected with a DNA-binding mutant (DBM) of RBPJκ (pRBPJκ-DBM) (Fig. 7g). Strikingly, the 4xCSL-luciferase activity was significantly enhanced in the present of

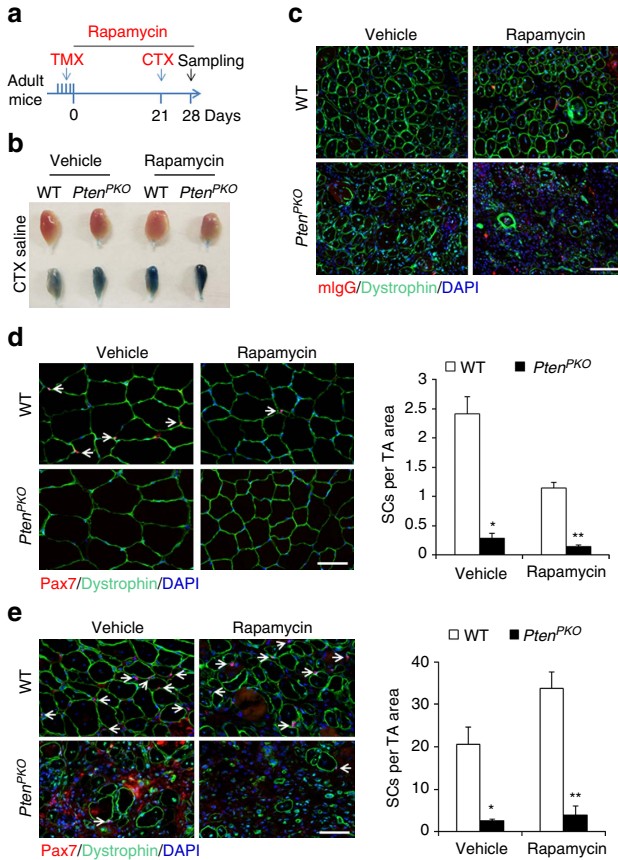

**Figure 6 | Inhibition of mTOR fails to rescue *Pten* KO induced SC depletion.** (**a**) Schematics showing experimental design (rapamycin inhibits mTOR). (**b**) Representative images of Evans blue dye labelled TA muscles 7 days after CTX injury. (**c**) Immunofluorescence of mIgG and dystrophin on TA muscle cross-sections of WT and *Pten^PKO* mice 7 days after CTX injury. Scale bar, 50 μm. (**d**) Left panel, immunofluorescence of Pax7 and dystrophin on uninjured TA muscle cross-sections. White arrow indicates SCs. Scale bar, 50 μm. Right panel, quantification of Pax7+ SC number. Data represent mean ± s.e.m. (*t*-test: *P<0.05; **P<0.001; Vehicle, n = 3; Rapamycin, n = 5, each group). (**e**) Left panel, immunofluorescence of Pax7 and dystrophin on injured TA muscle cross-sections. White arrow indicates SCs. Scale bar, 50 μm. Right panel, quantification of the number of SCs. Data represent mean ± s.e.m. (*t*-test: *P<0.05; **P<0.01; n = 3, each group).

FoxO1-ADA (Fig. 7g). These results demonstrate that nuclear FoxO1 enhances Notch signalling through interaction with RBPJκ. Therefore, cytoplasmic translocation of FoxO1 in *Pten*-null SCs should dampen Notch signalling, leading to impairments in SC maintenance.

**Notch activation prevents depletion of *Pten*-null SCs.** To directly test if inhibition of Notch signalling is responsible for the SC depletion in *Pten^PKO* mice, we examined whether activation of Notch signalling could rescue the depletion of *Pten*-null SCs. We used *Pax7^CreER* to drive concomitant *Pten* KO and Notch1 activation (*N1ICD* overexpression), abbreviated as *Pten^PKO/N1ICD* mice. We then examined the number of SCs on freshly isolated myofibers from these mice 28 days after TMX induction (Fig. 8a). Strikingly, the SC number was completely restored in *Pten^PKO/N1ICD* mice, compared with a 90% reduction of SCs in the *Pten^PKO* mice (Fig. 8b). Even

though MyoD+ SCs were still detectable in *Pten^PKO/N1ICD* mice 7 days after TMX induction (Fig. 8c), the total numbers of MyoD+ and MyoG+ cells were markedly reduced in *Pten^PKO/N1ICD* mice compared with the *Pten^PKO* mice (Fig. 8d,e). These results provide compelling genetic evidence that *Pten* KO leads to the depletion of SCs through inhibition of Notch signalling (Fig. 8f).

To distinguish whether N1ICD restores *Pten*-null SCs through stimulating their proliferation or preventing differentiation, we examined cell proliferation using Ki67 labelling. No significant differences were found in the abundance of Ki67+ SCs in uninjured TA muscles of WT, *Pten^PKO* mice and *Pten^PKO/N1ICD* mice 7 days after TMX induction (Supplementary Fig. 10a). We next examined the number of SCs in regenerating TA muscles 21 days after TMX treatment (Supplementary Fig. 10b) and found a remarkable increase of SCs in the TA muscles of *Pten^PKO/N1ICD* mice (∼185 SCs per TA area), which was 14-fold and 56-fold more than WT (∼13.2 SCs per TA area) and *Pten^PKO* (∼3.3 SCs per TA area) mice, respectively (Supplementary Fig. 10c). However, Ki67 staining revealed the percentage of proliferating SCs was decreased in *Pten^PKO/N1ICD* compared with WT mice (Supplementary Fig. 10d), suggesting that the increase of SCs was due to inhibition of differentiation rather than increased proliferation. Consistent with this notion, CTX-treated muscles of *Pten^PKO/N1ICD* mice regenerated extremely poorly, manifested by reduced muscle size and number of regenerated myofibers, but increased fibrosis (Supplementary Fig. 10e,f). These results are consistent with previous reports that Notch activation promotes the self-renewal but inhibits the differentiation of SCs during regeneration[12]. Collectively, although activation of Notch signalling blocks *Pten* deletion-induced SC depletion, it impairs SC function owing to inhibition of myogenic differentiation.

## Discussion

A distinctive feature of adult stem cells is their existence in a quiescent state with reversible mitotic arrest and low metabolic activity. In response to injury, stem cells are activated to proliferate and the proliferating cells subsequently withdraw from cell cycle either to differentiate or to self-renew. The self-renewed cells re-adopt quiescent state after regeneration to maintain long-term tissue homoeostasis[1,6]. How adult stem cells maintain quiescent state remains poorly understood. Our present study identifies an indispensable role of Pten in maintaining skeletal muscle stem cell quiescence. Specifically, we found that *Pten*-deficient SCs spontaneously exit from quiescent state, and undergo terminal differentiation without proliferation, leading to depletion of the SC pool and defective muscle regeneration in response to injury.

Our observation that Pten is predominantly expressed in SCs but not differentiated myotubes is consistent with the annotated expression pattern of Pten in published databases[44,45]. This expression pattern of Pten explains the lack of obvious phenotype in mice with myofiber-specific *Pten* KO[33,34]. Previous studies reported that conditional deletion of *Pten* in adult neural stem cells promotes their self-renewal by modulating G0–G1 cell cycle entry without signs of exhaustion, leading to constitutive neurogenesis[29,46]. In addition, inactivation of *Pten* causes short-term expansion of HSCs leading to the hyper-proliferation of myeloid and T-lymphoid lineages and leukaemogenesis[30,32]. Our results showing that *Pten* deletion in QSCs leads to its spontaneous activation and differentiation, resulting in rapid exhaustion of muscle stem cell pool are somehow different from the reported function of Pten in other stem cell types. The novel function of Pten in muscle

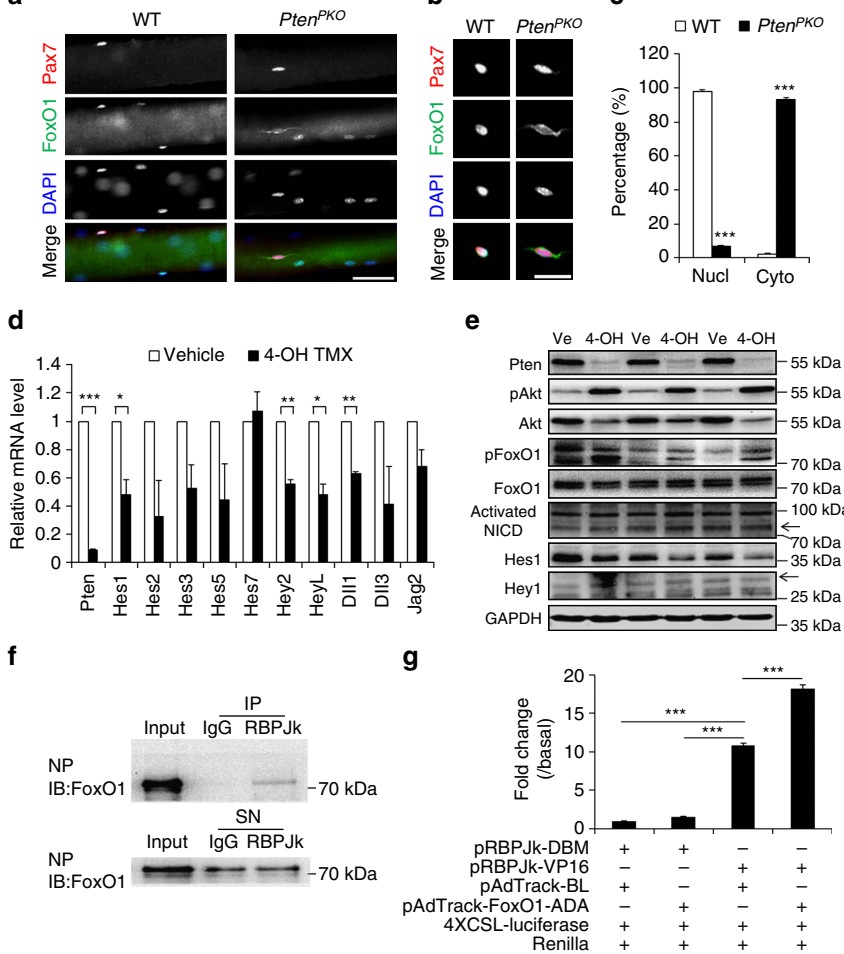

**Figure 7 | Pten KO inhibits Notch signalling by promoting cytoplasmic translocation of FoxO1.** (**a**) Immunofluorescence of Pax7 and FoxO1 on freshly isolated EDL myofibers from WT and Pten[PKO] mice 4 days after TMX induction. Scale bar, 20 μm. (**b**) Immunofluorescence of Pax7 and FoxO1 in SCs dissociated from EDL myofibers. Scale bar, 10 μm. (**c**) Quantification of the percentages of SCs with nuclear or cytoplasmic FoxO1 localisation. Data represent mean ± s.e.m. (t-test: ***P < 0.001; n = 3, each group; 20 myofibers per animal). (**d**) qRT-PCR analysis of relative mRNA levels of Notch related genes. Data are shown as mean ± s.e.m. (t-test: *P < 0.05, **P < 0.01, ***P < 0.001; n = 3, each group). (**e**) Western blot analysis of relative protein levels of Notch targets in Pax7[CreER]::Pten[f/f] primary myoblasts treated with vehicle or 4-OH-TMX. Arrows indicate the predicted band sizes. (**f**) Co-immunoprecipitation of FoxO1 and RBPJk in N1ICD[f/f] primary myoblasts co-infected with AdCre and AdFoxO1-ADA adenovirus. NP, nuclear protein; SN, supernatant after Co-IP. (**g**) Dual luciferase reporter assay in HEK293A cells co-transfected with pRBPJκ-DBM, pRBPJκ-VP16, pAdTrack-BL or pAdTrack-FoxO1-ADA plasmid with 4XCSL-Luciferase and Renilla plasmid. Data represent mean ± s.e.m. (t-test: ***P < 0.001; n = 5, each group).

stem cells thus adds to the repertoire of tissue type specific functions of Pten in adult stem cells.

Previous studies have confirmed the absolute requirement of Pax7-expressing SCs for regeneration of adult skeletal muscle, although other cell types with myogenic potential have been identified in muscle tissue[2,3,8,47–52]. Our data support the current knowledge that SCs are indispensable for muscle regeneration. We observed that ∼50% remnant of SCs induced by Pten deletion sufficiently maintain muscle regeneration. However, when the SC population drops to 10% or less, muscle regeneration fails. Our observations are consistent with previous reports on Pax7[CreER/+] and Pax7[CreERT2/+] genetic mouse models[2,3,47,48,50], suggesting that a threshold number of SCs are required for efficient muscle regeneration. Despite the regenerative defect observed in our Pten deletion mice, we cannot rule out the possibility that this regenerative defect would be overcome after a long recovery period due to the myogenic potential of other non-SC types[51,52].

Sporadic fusion of SCs into myofibers occurs in resting skeletal muscles, with or without cell cycle entry[53,54]. Consistently, we observed both EdU+ MyoG+ and EdU− MyoG+ cells in resting muscles of WT mice. Loss of Pten has been demonstrated to promote cell cycle entry and cell division[29,30,32,46]. Our observation that the majority of Pten-null SCs entered S-phase was consistent with previous studies. However, we found that Pten-null SCs undergo committed differentiation without exhibiting obvious cell proliferative or mitotic activity. Our data indicate that Pten-deficient SCs may undergo cell cycle arrest during the transition from S-phase to M-phase, and subsequently exit from the cell cycle and differentiate. Supporting our notion, studies on lower eukaryotes demonstrate that terminal differentiation of ectodermal epithelial stem cells can occur in G2-phase without requiring mitosis[55,56]. It would be interesting in the future to decipher how Pten deletion orchestrates the cell cycle exit and promotes differentiation.

As one of the downstream signalling of Akt pathway, mTORC1 has been found to be a master regulator controlling stem cell

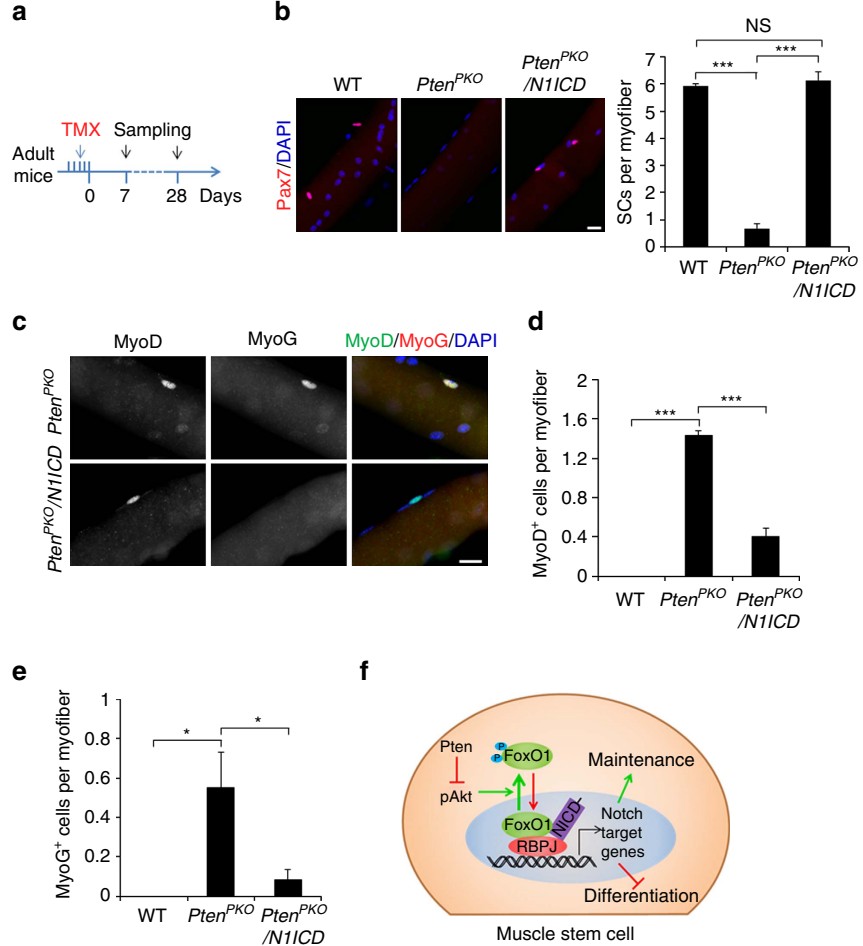

**Figure 8 | Activation of Notch signalling prevents depletion of *Pten*-null SCs.** (**a**) Schematics showing timing of TMX induction and sample collection. (**b**) Left panel, immunofluorescence of Pax7 on freshly isolated EDL myofiber of WT, *Pten^PKO* and *Pten^PKO/N1ICD* mice 28 days after TMX induction. Scale bar, 20 µm. Right panel, quantification of the number of SCs per EDL myofiber. Data represent mean ± s.e.m. (*t*-test: \*\*\**P* < 0.001; NS, no significance; *n* = 7, each group; 20 myofibers per animal). (**c**) MyoD and MyoG staining on freshly isolated EDL myofiber of *Pten^PKO* and *Pten^PKO/N1ICD* mice 7 days after TMX induction. Scale bar, 20 µm. (**d,e**) Quantification of the numbers of MyoD^+ (**d**) and MyoG^+ (**e**) cells per EDL myofiber. Data represent mean ± s.e.m. (*t*-test: \**P* < 0.05; \*\*\**P* < 0.001; *n* = 4, each group; 20 myofibers per animal). (**f**) A model depicting how Pten regulates SC maintenance and differentiation.

growth and metabolism[39,57–59]. Previous studies demonstrated that activation of mTOR signalling mediated the epidermal stem cell exhaustion induced by Wnt signalling[60], whereas pharmacological or genetically inactivation of mTOR by rapamycin can rescue *Pten* deletion-induced tumorigenesis and depletion of normal HSCs[32,37,38]. These reports indicate that dysfunction of *Pten*-deficient stem cells is dependent on Akt/mTOR pathway. However, despite the activation of mTORC1/S6 pathway observed in *Pten*-deficient SCs, we found that rapamycin administration fails to restore the SC pool, suggesting that SC depletion induced by *Pten* KO is independent of the mTOR pathway. One recent study reported that activation of mTORC1 is necessary and sufficient for the adaptive transition of quiescent muscle stem cells from G0 to G_Alert state, which puts stem cells in an alerted state under conditions of injury and stress[61]. Interestingly, mTORC1 activation induced by *Tsc1* deletion did not lead to SC depletion[61]. This finding supports our results that depletion of *Pten*-null SCs is not due to the observed activation of mTORC1 pathway. Instead, mTOR activation is probably responsible for the direct differentiation of SCs after their activation.

Our experiments demonstrate that Pten KO and OE down- or upregulated, respectively, the expression of Notch downstream genes in primary myoblasts. Subsequently, activation of Notch

signalling successfully restored the SC pool in the *Pten^PKO* mice. These results indicate that the modulation of Notch signalling by Pten pathway is an essential mechanism that enables Pten to regulate quiescence of SCs. Supporting our conclusion, SC-specific *RBP-J* KO phenocopies our *Pten^PKO* mice[13,14]. Mechanistically, *Pten* KO enhances Akt phosphorylation, which suppresses Notch signalling by inducing cytoplasmic translocation of FoxO1 (Fig. 8f). A recent study reported that FoxO3 promotes quiescence of adult muscle stem cells by upregulating Notch expression[41]. Interestingly, reduced levels of Notch signalling in *FoxO3*-deficient SCs did not deplete the SC pool in resting muscle as seen in both *RBP-J* and *Pten* deficient SCs. These results suggest that multiple regulators may mediate the interaction between Pten and Notch signalling in quiescent muscle stem cells. In addition to FoxO1, other downstream targets of Pten such as Akt and GSK3β have both been shown to modulate Notch signalling[43,62,63]. The cross-talk between Pten and Notch signalling extends our understanding of the complexity of intrinsic networks involved in stem cell quiescence.

## Methods
**Mice.** All mouse strains were obtained from Jackson Laboratory (Bar Harbor, ME) under the following stock numbers: *Pax7^CreER* (#012476), *Pten^f/f* (#006440) and

$Rosa^{Notch1}$(#008159). Mice were genotyped by PCR of ear DNA using genotyping protocols described by the supplier. The genotypes of experimental KO and associated control animals are as follows: $Pten^{PKO}$ ($Pax7^{CreER}$; $Pten^{f/f}$) and wild type ($Pten^{f/f}$), $Pten^{PKO}$/NICD ($Pax7^{CreER}$; $Pten^{f/f}$; $Rosa^{Notch1/+}$) and wild type ($Pten^{f/f}$; $Rosa^{Notch1/+}$). Mice were housed and maintained in the animal facility with free access to standard rodent chow and water. All procedures involving mice were approved by the Purdue University Animal Care and Use Committee. If not stated differently, 2- to 4-month-old mice were used for all experiments. Male or female mice were used and always gender-matched for each specific experiment.

**In vivo treatment.** TMX (Calbiochem) was prepared in corn oil at a concentration of $10 \, mg \, ml^{-1}$, and experimental and control mice were injected intraperitoneally with 2 mg TMX per day per 20 g body weight for 5 days to induce Cre-mediated deletion. TMX injections were initiated on adult mice, and experimental mice were used at the time stated in the text. In continuous labelling experiments, (EdU, Carbosynth) was administrated uninterruptedly to mice through drinking water ($0.3 \, mg \, ml^{-1}$) beginning at the onset of TMX induction. Drinking bottles were protected from light and fresh EdU-containing water was replaced every 3 days. Rapamycin (Calbiochem) was administered by IP injection at the dose of 4 mg per kg body weight per day for the indicated period. For preparation, rapamycin was first dissolved in ethanol at $10 \, mg \, ml^{-1}$ and diluted in 5% Tween-80 (Sigma) and 5% PEG-400 (Hampton Research) before injection[32].

**Muscle injury and regeneration.** Muscle regeneration was induced by CTX injection. Adult mice were anaesthetised using a ketamine–xylazine cocktail and CTX was injected (50 µl of 10 µM solution, Sigma) into TA muscle. Muscles were then harvested at the stated time to assess the completion of regeneration and repair.

**Single myofiber isolation and culture.** Single myofibers were isolated from EDL muscles of adult mice[64]. In brief, EDL muscles were dissected carefully and subjected to digestion with collagenase I ($2 \, mg \, ml^{-1}$, Sigma) in Dulbecco's Modified Eagle's Medium (DMEM, Sigma) for 1 h at 37 °C. Digestion was stopped by carefully transferring EDL muscles to a pre-warmed Petri dish (60-mm) with 6 ml of DMEM and single myofibers were released by gently flushing muscles with large bore glass pipette. Released single myofibers were then transferred and cultured in a horse serum-coated Petri dish (60-mm) in DMEM supplemented with 20% fetal bovine serum (FBS, HyClone), $4 \, ng \, ml^{-1}$ basic fibroblast growth factor (Promega), and 1% penicillin–streptomycin (HyClone) at 37 °C for indicated days.

**Primary myoblast culture and differentiation.** Primary myoblasts were isolated from hind limb skeletal muscles of wild type mice at the age of 4–6 weeks. Muscles were minced and digested in type I collagenase and Dispase B mixture (Roche Applied Science). The digestions were stopped with F-10 Ham's medium containing 20% FBS. Cells were then filtered from debris, centrifuged and cultured in growth medium (F-10 Ham's medium supplemented with 20% FBS, $4 \, ng \, ml^{-1}$ basic fibroblast growth factor, and 1% penicillin–streptomycin) on collagen-coated cell culture plates at 37 °C, 5% $CO_2$. For in vitro genetic deletion, 4-OH TMX (0.4 µM, Calbiochem) was added in culture medium for 2 days to induce Cre-mediated deletion.

For differentiation, primary myoblasts were seeded on BD Matrigel-coated cell culture plates and induced to differentiate in differentiation medium (DMEM supplemented with 2% horse serum and 1% penicillin-streptomycin).

**Adenovirus-mediated overexpression.** The adenovirus was generated using the AdEasy system[65]. The Pten ORF was cloned from cDNA of mouse primary myoblasts. The coding sequence of FoxO1-ADA was cloned from pCMV-FLAG-FoxO1-ADA plasmid, a gift from Dr Domenico Accili (Addgene plasmid #12149)[66]. Cloned sequence was inserted into the pAdTrack-CMV plasmid. The formed pAdTrack-Pten and pAdTrack-FoxO1-ADA (pAdTrack-CMV as the control) plasmid were digested by PmeI (NEB), and then transfected into DH5a-competent cell with pAdEasy-1. The positive recombinant plasmid was detected by PacI (NEB) digestion. For adenovirus generation, HEK293A cells (60–70% confluent) in 10-cm culture dishes were transfected with 4 µg of PacI-digested recombinant plasmid using Lipofectamine 2000 (Life Technologies) according to the protocol of the manufacturer. After 10 days of transfection, the cells were collected and recombinant adenovirus was released by freeze-thaw method. To increase the titres of adenovirus, two more rounds of infection were adapted to amplify the recombinant virus, and the titres were then determined by the expression of GFP. To analyse the function of Pten and FoxO1-ADA in primary myoblasts, adenovirus were added into myoblast growth medium for 12 h and then myoblasts were cultured in virus-free growth medium for another 2 days.

**Hematoxylin–eosin and immunofluorescence staining.** Whole muscle tissues from the WT and $Pten^{PKO}$ mice were dissected and frozen immediately in Optimal cutting temperature compound (OCT compound). Frozen muscles were cross sectioned (10 µm) using a Leica CM1850 cryostat. For hematoxylin and eosin staining, the slides were first stained in hematoxylin for 30 min, rinsed in running tap water and then stained in eosin for 1 min. Slides were dehydrated in graded ethanol and Xylene, and then covered using Permount.

For immunofluorescence staining, cross-sections, single myofibers or cultured cells were fixed in 4% PFA in PBS for 10 min, quenched with 100 mM glycine for 10 min, and incubated in blocking buffer (5% goat serum, 2% bovine serum albumin, 0.1% Triton X-100 and 0.1% sodium azide in PBS) for at least 1 h. Samples were then incubated with primary antibodies diluted in blocking buffer overnight at 4 °C. After washing with PBS, the samples were incubated with secondary antibodies and DAPI for 1 h at room temperature. Antibodies used for immunofluorescence staining were listed in Supplementary Table 1. For EdU staining, samples with EdU incorporation were first fixed in 4% PFA in PBS for 10 min. EdU was visualized by Click-iT method[67] with red fluorescent dye tetramethylrhodamine azide (Invitrogen). Samples were then subjected to Pax7 immunofluorescence staining.

All hematoxylin and eosin staining images were captured using a Nikon D90 digital camera mounted on a microscope with a × 20 objective. All immunofluorescent images were captured using a Leica DM 6000B microscope with a × 20 objective, or Zeiss LSM 700 Confocal with a × 20 objective. Images for WT and KO samples were captured using identical parameters. The number of regenerated myofibers per $mm^2$, average cross-section area of regenerated fibres, and percentage of regenerated area were calculated by PhotoShop software. All images shown are representative results of at least three biological replicates.

**In situ TUNEL assay to detect cell apoptosis.** For TUNEL and Pax7 staining, slides were fixed in 4% PFA for 10 min and then subjected to the TUNEL reaction using the CF488A TUNEL Assay Apoptosis Detection Kit (Biotium) according to the manufacturer's instructions. For negative control, samples were added TUNEL reaction buffer without TdT Enzyme. Samples treated with DNaseI for 30 min before TUNEL staining was set up as positive ccontrol. Counterstaining of Pax7 was then performed as regular immunofluorescence staining procedure.

**Total RNA extraction and Real-time PCR.** Total RNA was extracted from tissues using TRIzol reagent according to the manufacturer's instructions. RNA was treated with RNase-free DNase I to remove contaminating genomic DNA. The purity and concentration of the total RNA were determined by a spectrophotometer Nanodrop 2000c (Thermo Fisher). 3 µg of total RNA was reverse transcribed using random primers with M-MLV reverse transcriptase (Invitrogen). Real-time PCR was carried out in a Roche Light Cycler 480 PCR System with SYBR Green Master Mix and gene-specific primers were listed in Supplementary Table 2. The $2^{-\Delta\Delta Ct}$ method was used to analyse the relative changes in each gene's expression normalized against 18S rRNA expression.

**Protein extraction and western blot analysis.** Total protein was isolated from cells using RIPA buffer containing 25 mM Tris-HCl (pH 8.0), 150 mM NaCl, 1 mM EDTA, 0.5% NP-40, 0.5% sodium deoxycholate and 0.1% SDS. Protein concentrations were determined using Pierce BCA Protein Assay Reagent (Pierce Biotechnology). Proteins were separated by SDS-PAGE, transferred to a polyvinylidene fluoride membrane (Millipore Corporation), blocked in 5% fat-free milk for 1 h at room temperature and then incubated with primary antibodies in 5% milk overnight at 4 °C. Membrane was then incubated with secondary antibody for 1 h at room temperature. Antibodies used for western blot analysis were listed in Supplementary Table 3. Immunodetection was performed using enhanced chemiluminescence western blotting substrate (Santa Cruz Biotechnology) and detected with a FluorChem R System (Proteinsimple). Results shown in the figures are representative results from three independent experiments. Uncropped original blots are available in Supplementary Fig. 11.

**Co-immunoprecipitation assay.** Total or nuclear protein was extracted from $Rosa^{Notch1}$ primary myoblast infected with or without AdCre and AdFoxO1-ADA adenovirus based on method modified from Chi et al[68]. The lysate was precleared with protein A/G agarose at 4 °C for 2 h. Then 4 µg of primary antibody anti-RBPJκ (#H-50, Santa Cruz Biotechnology ) or rabbit IgG (#2729, Cell Signaling) was added into cell lysate contains 500 µg total protein, and rotating at 4 °C overnight. Protein A/G agarose was added into cell lysate and rotating for 2 h at 4 °C. The samples were washed extensively for six times, and subjected to western blot analysis.

**Luciferase assay.** HEK293A cells were seeded in 48-well plates 1 day before transfection. Cells were transfected with 0.3 µg DNA using Lipofectamine 2000 (Invitrogen) with 4xCSL-luciferase (a gift from Dr Raphael Kopan, Addgene plasmid #41726)[69], pRS2-RBPJκ-VP16, pRS2-RBPJκ-DBM (a gift from Dr Mark Mercola)[70], pAdTrack-BL or pAdTrack-FoxO1-ADA, and pRL-tk (Promega) to normalize transfection efficiency. Cells were harvested 48 h after

transfection and analysed with the Dual-Luciferase Reporter Assay System (Promega). Relative luciferase activity was presented as fold change compared with basal sample. HEK293A cell line was certified by ATCC for mycoplasma-free when cells were purchased, and cell identity was authenticated by morphological features.

**Statistical analysis.** Trial experiments or experiments done previously were used to determine sample size with adequate statistical power. The researchers involved in the *in vivo* treatments were not completely blinded, but all images were randomly captured from sample and analysed in a blinded manner. No data were excluded from following statistical analysis. All analyses were conducted with Student's *t*-test with a two-tail distribution. All experimental data are represented as mean ± s.e.m. Comparisons with *P* values < 0.05 were considered statistically significant.

**Data availability.** The data that support the findings of this study are available from the corresponding author on request.

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

## Acknowledgements

This work was supported by grants from the US National Institutes of Health (R01AR060652 to S.K. and P30CA023168 to Purdue University Center for Cancer Research), and Purdue incentive grant from Purdue University Office of Vice President for Research (OVPR) to S.K. We thank Dr Chen-Ming Fan (Carnegie Institute) for providing the Pax7$^{CreERT2}$ mice, Dr Mark Mercola (University of California, San Diego) for providing pRS2-RBPJκ-VP16 and pRS2-RBPJκ-DBM plasmids, Dr Domenico Accili (Columbia University Medical Center) for providing pCMV-FLAG-FoxO1-ADA plasmid, Dr Raphael Kopan (University of Cincinnati College of Medicine) for providing 4xCSL-luciferase plasmid, Dr YongXu Wang (University of Massachusetts Medical School) for generous present of Adenovirus-AdEasy overexpression system, Dr Xiaoqi Liu (Purdue University) for sharing reagents (antibodies), Jun Wu for mouse colony maintenance and members of the Kuang laboratory for valuable comments.

## Author contributions

F.Y. and S.K. conceived the project, designed the experiments, analysed the data and wrote the manuscript. F.Y., P.B., C.W., T.S. and Y.N. performed the experiments. T.R. and T.G. provided reagents.

## Additional information

**Competing financial interests:** The authors declare no competing financial interests.

