## [Peer Review File · Nature Communications]

Reviewers' comments:

Reviewer #1

Expert in PTEN signaling, stem cells

(Remarks to the Author):

PTEN was documented to be a key regulator in controlling the transition between quiescence and activation of adult stem cells via suppression of PI3K-Akt signaling. Whether and how PTEN plays a similar role in muscle stem cell is not well investigated. In this work, authors have used stage-specific KO of PTEN in quiescent, proliferating, and differentiating muscle stem cells and found that PTEN is critical in maintaining quiescent stem cells, in supporting self-renewal of active stem cells. Loss of PTEN led to depletion of quiescent stem cells, and reduced regeneration capacity of activated stem cells. Authors further looked into the underlying mechanism, they found that while the mTOR activity is increased upon PTEN deletion, however, inhibition of mTOR by Rapamycin cannot rescue the regeneration capacity of PTEN-KO stem cells. A novel finding is the link between PTEN-Akt-Foxo1 with Notch pathway as suggested in the context of muscle regeneration. Major concern, authors need to test whether Foxo1 can directly bind RBP-J and regulate Notch target gene expression in muscle stem cells.

Reviewer #2

Expert in muscle regeneration

(Remarks to the Author):

This interesting paper analyzes the effects of deletion of PTEN in muscle satellite cells, and shows clearly that in the absence of this factor satellite cells differentiate spontaneously without entering cell division.

This is important as PTEN plays very different roles in stem cells from tissues with high homeostatic turnover. Generally speaking, the work is high quality and the topic worthy of publication in this journal.

While this result is well documented and solid, it is unclear why the authors choose to interpret this as an effect on self-renewal. Clearly, if a cell differentiates it can no longer self-renew but this is also the case if a cell dies and one would hardly call a survival defect "lack of self-renewal". This may be semantic, but in my opinion that the slant given to this paper is going to end up confusing the field rather than clarifying it. Indeed, if lack of PTEN affected self-renewal specifically, the depletion of SC following PTEN deletion would be much slower reflecting the low rate self-renewal of these progenitors in homeostatic conditions.

A number of outstanding questions are left that should be addressed.

While the efficiency of Pax7 CT2 can be very high, it has been shown that un-recombined SC will eventually repopulate muscles unless TAM treatment is repeated. Would this take place here and if not, how do the authors interpret this? Could lack of PTEN have non-cell autonomous effects?

Figure 3 shows a defect in regeneration in mice 21 days after Tam treatment, but it does not investigate the effects on SC. Figure 4 shows a defect in SC following damage at 7, 14 and 21 days. If the SC are already depleted at day 21 post TAM independent of damage as shown in figure 2, why do the authors feel the need to do three rounds of damage in figure 4? This experiment does not teach us much. An analysis of SC following an early round of damage (e.g. 4 days after Tam) would be required to conclude whether damage-induced activation leads to a faster depletion of PTEN KO SC.

The authors show that Notch targets are down in PTEN deleted SC, and present a plausible

mechanism involving FoxO1. They also present evidence that transgenic activation of the notch pathway rescues PTEN KO SC by blocking their differentiation. However, their analysis of this experiment should include an assessment of proliferation. NCID overexpression is well known to block differentiation even in wild type cells and to induce their proliferation. Thus NCID may be dominant over lack of PTEN but hardly restore normal SC dynamics, essentially acting as an oncogene.

The "muscle recovery rate" is not a standard measurement and yet is not defined.

Minor: It would be interesting to assess the fate of the differentiating SC progeny. Do they fuse into fibers? However this would required breeding a transgenic marker that can be activated by CRE in the Pax7CT2 mice, and it is not critical to the main message.

Reviewer #3

Expert in muscle regeneration

(Remarks to the Author):

Review for NCOMMS-16-08598

Satellite cell-specific depletion of Pten leads to the loss of satellite cells in skeletal muscle tissue, preventing regeneration. Mechanistically, the authors show that Pten loss activates Foxo1, inhibiting Notch signaling leading to loss of quiescence in satellite cells. The authors conclude that Pten loss induces differentiation of satellite cells without an intervening cell cycle. Thus, the authors assert that Pten is essential to maintain satellite cells in quiescence and required for self-renewal.

Overall the experiments support the authors conclusions. The mechanistic data are convincing and support the conclusions made by the authors. However, the data supporting the claim that Pten null satellite cells differentiate as opposed to simply undergoing cell death requires additional experimental data for support. An alternative explanation for the authors data is that Pten is required for satellite cell survival. Loss of Pten leads to satellite cell death and the observed phenotypes. The authors have attempted to refute this possibility by analyzing apoptosis and by performing EdU labeling experiments. Unfortunately these experiments are somewhat equivocal and there are technical concerns. First, the half-life of EdU(BrdU) in vivo is an hour or less (Barker, JM et al.. PLoS ONE 8: e63692. doi: 10.1371/journal.pone.0063692.) and EdU injections at once per day as performed by the authors fails to label the proliferating satellite cells. To label the satellite cells, the authors need to add EdU to the drinking water as it is not possible to perform a sufficient number of injections to label the satellite cell population. This could easily explain the similarities in EdU labeling between the wild type and Pten ko satellite cells seen in Fig. 2h and Fig. S2.

If Pten null satellite cells are differentiating then the numbers of centrally located nuclei should dramatically increase. No data are presented that convincingly demonstrate that the loss of Pten null satellite cells increases the numbers of myonuclei, particularly centrally located myonuclei. Furthermore, the Caspase counts are not an appropriate method for scoring apoptotic cells as Caspase 3 activation is required for myogenesis (Dick, SA et al., Proceedings of the National Academy of Sciences 112: 52. doi: 10.1073/pnas.1512869112). More data are needed to convincingly demonstrate that Pten ko cells are differentiating as opposed to dying.

A conceptual issues noted by the authors is the "surprising" result that satellite cells are undergoing differentiation without an intervening cell cycle. The authors need to revise the text and cite the recent references that have demonstrated by lineage labeling that satellite cells fuse into myofibers without an intervening cell cycle (Keefe, AC et al. 2015. Muscle stem cells contribute to myofibres in sedentary adult mice.. Nat Commun 6: 7087. doi:

10.1038/ncomms8087; Pawlikowski, B. Et al.,2015. Pervasive satellite cell contribution to uninjured adult muscle fibers. *Skeletal Muscle* 5: doi: 10.1186/s13395-015-0067-1. <http://dx.doi.org/10.1186/s13395-015-0067-1>).

Finally, no data are presented to show that Pten is required for self-renewal and all text that asserts Pten is required for self-renewal should be omitted. If Pten is required for maintaining quiescence as posited by the authors, loss of Pten will result in loss of satellite cells. This does not establish that Pten plays a role in self-renewal.

The following minor/technical issues also need to be addressed:

Fig. 1: The percentage of satellite cells that are Pten+ requires quantification in freshly isolated myofibers and in tissue sections. (The latter could be added to figure 2).

Insets at higher magnification identifying satellite cells are needed for Fig. 2c. The nearly complete lack of MyoD+ and MyoG+ satellite cells per myofiber in Fig. 2e and 2f, respectively is unusual and not in agreement with prior published data. Why are no MyoD+ or MyoG+ cells present in wild type myofiber cultures? The EdU labeling is not possible to see in Fig. 2h (see comments above as well for EdU).

Supp Fig. 2a the wild type controls need to be shown along with the Pten ko sections.

The authors have identified a potentially important regulatory pathway to maintain quiescent satellite cells. However, the data presented do not sufficiently distinguish between a requirement of Pten for cell survival as opposed to a requirement for Pten to maintain quiescence. Additional data will support the authors' conclusions and provide an important contribution to the field.

Reviewers' comments:

Authors' Response: We thank the Editors and Reviewers for their insightful comments and hope the following point-to-point responses will address their concerns. All revisions have been highlighted with yellow color in the revised manuscript.

Reviewer #1

Expert in PTEN signaling, stem cells
(Remarks to the Author):

PTEN was documented to be a key regulator in controlling the transition between quiescence and activation of adult stem cells via suppression of PI3K-Akt signaling. Whether and how PTEN plays a similar role in muscle stem cell is not well investigated. In this work, authors have used stage-specific KO of PTEN in quiescent, proliferating, and differentiating muscle stem cells and found that PTEN is critical in maintaining quiescent stem cells, in supporting self-renewal of active stem cells. Loss of PTEN led to deletion of quiescent stem cells, and reduced regeneration capacity of activated stem cells. Authors further looked into the underlying mechanism, they found that while the mTOR activity is increased upon PTEN deletion, however, inhibition of mTOR by Rapamycin cannot rescue the regeneration capacity of PTEN-KO stem cells. A novel finding is the link between PTEN-Akt-Foxo1 with Notch pathway as suggested in the context of muscle regeneration.

Major concern, authors needs to test whether Foxo1 can directly bind RBP-J and regulate Notch target gene expression in muscle stem cells.

Authors' Response: We appreciate the reviewer's expert comments. As the reviewer suggested, we tested whether FoxO1 directly binds RBPJ to regulate Notch target gene expression in primary myoblasts. We first found overexpression of a constitutively active FoxO1 mutant (FoxO1-ADA) in primary myoblasts upregulated the expression of several Notch target genes (Supplementary Fig. 9b,c). Moreover, we detected the endogenous interaction of FoxO1 and RBPJk in primary myoblasts (Supplementary Fig. 9d), and the interaction was significantly enhanced by activation of Notch signaling and overexpression of FoxO1-ADA (Fig. 7f). To confirm the interaction of FoxO1-RBPJk on Notch target gene expression, we further measured RBPJk induced luciferase activity in the presence or absence of FoxO1-ADA in HEK293A cells. Transfection of a constitutively active form of RBPJk made by fusion with VP16 transactivation domain (pRBPJk-VP16) dramatically increased the 4xCSL-luciferase activity compared to the control transfected with a DNA-binding mutant (DBM) of RBPJk (pRBPJk-DBM) (Fig. 7g). Strikingly, the 4xCSL-luciferase activity was significantly enhanced in the present of FoxO1-ADA (Fig. 7g). Consistent with previous reports^{1,2,3}, our results demonstrated that FoxO1 regulates Notch signaling by interaction with RBPJk in myogenic progenitors. We hope these results will satisfy the reviewer.

Reviewer #2

Expert in muscle regeneration
(Remarks to the Author):

This interesting paper analyzes the effects of deletion of PTEN in muscle satellite cells, and shows clearly that in the absence of this factor satellite cells differentiate spontaneously without entering cell division. This is important as PTEN plays very different roles in stem cells from tissues with high homeostatic turnover. Generally speaking, the work in high quality and the topic worthy of publication in this journal.

While this result is well documented and solid, it is unclear why the authors choose to interpret this as an effect on self-renewal. Clearly, if a cell differentiates it can no longer self renew but this is also the case if a cell dies and one would hardly call a survival defect "lack of self renewal". This may be semantic, but I am the opinion that the slant given to this paper is going to end up confusing the field rather than clarifying it. Indeed, if lack of PTEN affected self renewal specifically, the depletion of SC following PTEN deletion would be much slower reflecting the low rate self-renewal of these progenitors in homeostatic conditions.

Authors' Response: We agree with the reviewer's comments. Our data indicate that *Pten* is required for quiescence maintenance of muscle stem cells and loss of *Pten* leads to terminal differentiation. To clarify this finding in our manuscript and avoid confusion, we have reworded most of "self-renewal" and rewritten the related sections as suggested by the reviewer.

A number of outstanding questions are left that should be addressed.

1. While the efficiency of Pax7 CT2 can be very high, it has been shown that un-recombined SC will eventually repopulate muscles unless TAM treatment is repeated. Would this take place here and if not, how do the author interpret this? Could lack of PTEN have non-cell autonomous effects?

Authors' Response: In brief, in contrast to what was reported in *Pax7^{CreERT2/fi}* mice⁴, we did not observe the repopulation of un-recombined SCs in our study. Our conclusion is based on multiple lines of evidence as detailed below.

1) Despite 2% of SCs seems to have escaped *Pten* deletion at Day 7 after tamoxifen induction in *Pten^{PKO}* mice, the number of SCs remained very low during regeneration and never caught back, no matter in the muscles injured at Day 4 or Day 21 after tamoxifen induction (Supplementary Fig. 1e and Fig. 2b).

2) To test whether these remnant *Pten⁺* SCs in *Pten^{PKO}* mice could restore the SC pool after long term tamoxifen withdraw and repopulate injured muscle. We examined the number of SC in uninjured TA muscles 4 months after tamoxifen induction (Supplementary Fig. 3a). Notably, the number of Pax7⁺ SCs remained low in *Pten^{PKO}* mice, only ~3% of the level in WT mice (0.06 vs 1.92/TA area, Supplementary Fig. 3b). With the few number of SCs, the muscles of *Pten^{PKO}* mice regenerated extremely poor 14 days after duple CTX injury, revealed by reduced muscle size and the number of regenerated myofiber (Supplementary Fig. 3c,d). Moreover, the number of Pax7⁺ SCs in injured TA muscle of *Pten^{PKO}* mice was only ~2% of that in WT mice (0.08 vs 3.62/TA area, Supplementary Fig. 3e). These results suggest that the remnant *Pten⁺* SCs fail to repopulate SC pool and repair injured muscles even after a long term recovery. It is possible that the number of the *Pten⁺* cells is too low to allow sufficient repopulation without proliferative senescence.

However, we cannot exclude the possibility that after a longer time of chasing, the injured muscles in *Pten^{PKO}* mice would be repaired eventually and catch up with the muscles in WT mice. It is possible that other non-satellite stem cell types, such as mesoangioblasts, Pw1⁺ interstitial cells, and other interstitial cell types^{5, 6, 7} may contribute to the regeneration of injured muscles. Due to the 3-month time frame allowed for revision, we were unable to perform longer term chasing.

The phenotypic differences of SCs observed in tamoxifen-induced *Pax7^{CreERT2/fi}* and ours *Pax7^{CreERT2}/Pten^{fi}* system could dependent on several complex factors^{4, 8, 9}. First, the target genes were different. The chromatin structure of different target genes could affect Cre activity and efficiency. Moreover, the time

of protein turnover would affect the subsequent phenotypes. Furthermore, *Pax7*-null cells seem to persist in the muscle while the *Pten*-null satellite cells are lost due to differentiation.

For the second part of the question, we are confident that the effect of *Pten* deletion on SCs is cell autonomous because our system only leads to *Pten* deletion in satellite cells in the muscle. Although some cells outside the muscle (for example in the brain^{10, 11}) may be targeted by *Pax7*^{Cre}, the influence of those cells on satellite cells has not been documented. Even if the *Pax7Cre* resulted in sporadic deletion of *Pten* in the myofiber (due to fusion of satellite cells), the possibility of non-cell autonomous effect would be extremely low because our expression analysis indicate that *Pten* is not expressed by myofibers (Fig. 1).

2. Figure 3 shows a defect in regeneration in mice 21 days after Tam treatment, but it does not investigate the effects on SC. Figure 4 shows a defect in SC following damage at 7, 14 and 21 days. If the SC are already depleted at day 21 post TAM independent of damage as shown in figure 2, why do the authors feel the need to do three rounds of damage in figure 4? This experiment does not teach us much. An analysis of SC following an early round of damage (e.g. 4 days after Tam) would be required to conclude whether damage-induced activation leads to a faster depletion of PTEN KO SC.

Authors' Response: This is a great comment that we divide into three parts (as underlined).

1) As the reviewer suggested, we examined the number of SCs in muscles that were injured at Day 21 after tamoxifen induction. At 7 days after CTX injury, the number of SCs in *Pten*^{PKO} mice was reduced by 88% compared to WT mice (Supplementary Fig. 2b). The results indicate that remnant SCs in *Pten*^{PKO} mice cannot repopulate SC pool during regeneration.

2) Our data showed that prior to depletion (Injury at Day 5 after tamoxifen), activated *Pten*-null SCs can regenerate the damaged muscles. The original purpose for the three rounds of injury was to further test whether these activated *Pten*-null SCs could replenish SC pool and repair damaged muscle with continuous regenerative cues. As the reviewer feels unnecessary to show these experiments, we have removed these results in the revised manuscript.

3) Indeed, in our original manuscript (Supplementary Fig. 3f), we analyzed the number of SCs in regenerating muscles when CTX was injected at Day 5 after tamoxifen induction. The number of *Pax7*⁺ cells was reduced by ~70% in the *Pten*^{PKO} compared to WT mice 7 days after injury (Supplementary Fig. 3f in original manuscript). As the reviewer suggested, we performed a new experiment in which CTX was injected at Day 4 after tamoxifen induction, to test whether early damage-induced activation of *Pten*-null SCs leads to their faster depletion (Supplementary Fig. 1). Consistent with our previous results, TA muscles of *Pten*^{PKO} and WT mice regenerated equally well 14 days after injury (Supplementary Fig. 1b-d), the number of *Pax7*⁺ cells was reduced by ~74% in the *Pten*^{PKO} compared to WT mice 14 days after injury, whereas ~78% of *Pax7*⁺ cells was decreased in uninjured muscles of *Pten*^{PKO} mice (Supplementary Fig. 1e). These results suggest the SC depletion occurs at similar rate with or without injury.

3. The authors show that Notch targets are down in PTEN deleted SC, and present a plausible mechanism involving FoxO1. They also present evidence that transgenic activation of the notch pathway rescues PTEN KO SC by blocking their differentiation. However, their analysis of this experiment should include an assessment of proliferation. NCID overexpression is well known to block differentiation even in wild type cells and to induce their proliferation. Thus NCID may be dominant over lack of PTEN but hardly restore normal SC dynamics, essentially acting as an oncogene.

Authors' Response: We agree with reviewer's suggestion. We have now analyzed the proliferation of SCs in *Pten*^{PKO} mice and *Pten*^{PKO}/*N1ICD* mice in both resting and regenerating muscles (Supplementary Fig. 10). In brief, we found that NICD rescued satellite cell depletion not because of increases in proliferation, but due to inhibition of differentiation (as the reviewer correctly pointed out that Notch inhibits differentiation). Also the inhibition of proliferation by NICD was reported in a previous study¹². Given the observation, we agree with the reviewer that NICD may have a dominant effect over Pten. Nevertheless, the rest of our study showing Pten gain- and loss-of-function up- and down-regulated Notch targets, respectively, provides direct evidence that Pten regulates Notch signaling. The analysis of FoxO1-Rbpj interaction and transcriptional regulation further strengthens our proposed mechanism.

4. The "muscle recovery rate" is not a standard measurement and yet is not defined.

Authors' Response: "muscle recovery rate" has been corrected to "Ratio of muscle weight (CTX/Control)" in the revised manuscript.

Minor: It would be interesting to assess the fate of the differentiating SC progeny. Do they fuse into fibers? However this would required breeding a transgenic marker that can be activated by CRE in the Pax7CT2 mice, and it is not critical to the main message.

Authors' Response: Good suggestion. We used EdU incorporation assay to demonstrate that the differentiating *Pten*-null SCs eventually fused into myofiber (Fig. 4g,h and Supplementary Fig. 8a). Also refer to detailed response to Comment 1 of the Reviewer #3).

Reviewer #3

Expert in muscle regeneration

(Remarks to the Author):

Review for NCOMMS-16-08598

Satellite cell-specific depletion of Pten leads to the loss of satellite cells in skeletal muscle tissue, preventing regeneration. Mechanistically, the authors show that Pten loss activates Foxo1, inhibiting Notch signaling leading to loss of quiescence in satellite cells. The authors conclude that Pten loss induces differentiation of satellite cells without an intervening cell cycle. Thus, the authors assert that Pten is essential to maintain satellite cells in quiescence and required for self-renewal.

1. Overall the experiments support the authors conclusions. The mechanistic data are convincing and support the conclusions made by the authors. However, the data supporting the claim that Pten null satellite cells differentiate as opposed to simply undergoing cell death requires additional experimental data for support. An alternative explanation for the authors data is that Pten is required for satellite cell survival. Loss of Pten leads to satellite cell death and the observed phenotypes. The authors have attempted to refute this possibility by analyzing apoptosis and by performing EdU labeling experiments. Unfortunately these experiments are somewhat equivocal and there are technical concerns. First, the half-life of EdU(BrdU) in vivo is an hour or less (Barker, JM et al.. PLoS ONE 8: e63692. doi: 10.1371/journal.pone.0063692.) and EdU injections at once per day as performed by the authors fails to label the proliferating satellite cells. To label the satellite cells, the authors need to add EdU to the

drinking water as it is not possible to perform a sufficient number of injections to label the satellite cell population. This could easily explain the similarities in EdU labeling between the wild type and Pten ko satellite cells seen in Fig. 2h and Fig. S2.

Authors' Response: We appreciate the reviewer's expert comments. We divide the comment into 2 parts (as underlined) to better address.

1) To provide more convincing evidence of cell death, we performed in situ terminal deoxynucleotidyl transferase dUTP nick-end labeling (TUNEL) assay, an unbiased method to detect DNA strand breaks in apoptotic cells. As positive control, TUNEL⁺ SCs were observed on myofibers treated with DNaseI (Supplementary Fig. 4a). However, TUNEL⁺ SCs were not identified on myofibers of WT (n=522 SCs) and *Pten*^{PKO} (n=521 SCs) mice 7 days after tamoxifen induction (Fig. 3a). Similar results were observed on TA muscle cross-sections (Supplementary Fig. 4b). These observations are consistent with our previous results shown by staining of cleaved-caspase3 in SCs. Therefore, depletion of *Pten*-null SCs is not due to cell death.

2) The reviewer was absolutely correct that EdU injections at once per day could under estimate the proliferating satellite cells due to the short half-time of EdU in vivo. As suggested by the reviewer, we performed a new EdU incorporation experiment by added the EdU in drink water (0.3mg/ml) to achieve continual labeling (Supplementary Fig. 6). We reexamined the cell cycle entry and proliferation of *Pten*-null SCs. In this case, EdU⁺ SCs were detected in both WT and *Pten*^{PKO} mice 7 days after tamoxifen induction. Despite fewer total SCs (Fig. 2b), significant more EdU⁺ SCs were found in *Pten*^{PKO} mice (0.82/myofiber vs 0.37/myofiber in WT, 20.2% vs 6.3% in WT) 7 days after tamoxifen induction (Fig. 4a-c). These results suggest that *Pten* KO promotes activation and S-phase entry of SCs. However, these SCs did not seem to go through the cell division since less than one EdU⁺ SCs was detected in each myofiber of *Pten*^{PKO} mice and we never observe any EdU⁺ SC doublets.

Notably, we found ~92% of the differentiating MyoG⁺ cells were EdU⁺ in *Pten*^{PKO} mice 7 days after tamoxifen induction (Fig. 4d-f). This finding enabled us to track the fate of these cells. To test whether these differentiating MyoG⁺ cells were eventually fused into myofiber, we performed immunostaining of EdU and fiber membrane protein dystrophin in TA muscle cross-sections. Strikingly, the number of EdU⁺ myonuclei was increased significantly in *Pten*^{PKO} (~8.7/TA area) compared to WT mice (~1.3/TA area) at 7 days after tamoxifen induction (Fig. 4g,h and Supplementary Fig. 8a), suggesting that the differentiating MyoG⁺ cells in *Pten*^{PKO} mice fused into myofiber. These results again support our conclusion that depletion of *Pten*-null SCs is due to differentiation.

2. If *Pten* null satellite cells are differentiating then the numbers of centrally located nuclei should dramatically increase. No data are presented that convincingly demonstrate that the loss of *Pten* null satellite cells increases the numbers of myonuclei, particularly centrally located myonuclei. Furthermore, the Caspase counts are not an appropriate method for scoring apoptotic cells as Caspase 3 activation is required for myogenesis (Dick, SA et al., Proceedings of the National Academy of Sciences 112: 52. doi: 10.1073/pnas.1512869112). More data are needed to convincingly demonstrate that *Pten* ko cells are differentiating as opposed to dying.

Authors' Response: Centralized myonuclei are a hallmark for regenerated myofibers, but not for the spontaneous fusion of SCs in resting muscles. We now provide evidence that the EdU-labeled cells did not become centrally localized after fusion (Fig. 4g and Supplementary Fig. 8a). Similar phenomenon was reported in recent study by Dr. Gabrielle Kardon group¹³.

Nevertheless, we enumerated central myonuclei as the reviewer suggested, but did not see any differences between WT and *Pten*^{PKO} mice (Supplementary Fig. 8c). Also, because *Pten*-null satellite cells differentiate without proliferation, it would be hard to detect any robust increases in myonuclei (in theory 6 myonuclei increase over ~260 total myonuclei per fiber). This was indeed what we saw after counting myonuclei number per fiber (Supplementary Fig. 8b).

We appreciate the reviewer's comments about role of Caspase-3 in myogenesis, and now performed TUNEL assay to validate our observation (Fig. 3a and Supplementary Fig. 4). The observation of EdU⁺ myonuclei confirmed that *Pten*-null SCs indeed differentiate and fuse into the host myofiber (Fig. 4g,h and Supplementary Fig. 8a).

3. A conceptual issues noted by the authors is the "surprising" result that satellite cells are undergoing differentiation without an intervening cell cycle. The authors need to revise the text and cite the recent references that have demonstrated by lineage labeling that satellite cells fuse into myofibers without an intervening cell cycle (Keefe, AC et al. 2015. Muscle stem cells contribute to myofibres in sedentary adult mice.. Nat Commun 6: 7087. doi: 10.1038/ncomms8087; Pawlikowski, B. Et al.,2015. Pervasive satellite cell contribution to uninjured adult muscle fibers. Skeletal Muscle 5: doi: 10.1186/s13395-015-0067-1. <http://dx.doi.org/10.1186/s13395-015-0067-1>).

Authors' Response: Thank you for the suggestion. We have revised this discussion and cited the recent studies by Keefe, AC et al.¹³ and Pawlikowski, B. Et al¹⁴.

4. Finally, no data are presented to show that Pten is required for self-renewal and all text that asserts Pten is required for self-renewal should be omitted. If Pten is required for maintaining quiescence as posited by the authors, loss of Pten will result in loss of satellite cells. This does not establish that Pten plays a role in self-renewal.

Authors' Response: We agree and have rewritten the relevant words/sections.

The following minor/technical issues also need to be addressed:

1. Fig. 1: The percentage of satellite cells that are Pten⁺ requires quantification in freshly isolated myofibers and in tissue sections. (The latter could be added to figure 2).

Authors' Response: In reality the Pten antibody works very well on immunostaining of myoblast and isolated myofibers, but not on muscle cross-sections. We therefore only presented the quantification of Pten⁺ satellite cells performed on isolated myofibers (Supplementary Fig. 1a).

2. Insets at higher magnification identifying satellite cells are needed for Fig. 2c. The nearly complete lack of MyoD⁺ and MyoG⁺ satellite cells per myofiber in Fig. 2e and 2f, respectively is unusual and not in agreement with prior published data. Why are no MyoD⁺ or MyoG⁺ cells present in wild type myofiber cultures? The EdU labeling is not possible to see in Fig. 2h (see comments above as well for EdU).

Authors' Response: As the reviewer suggested, we have added the image with higher magnification identifying satellite cells for Fig. 2c. We hope the reviewer recognizes that the immunostaining and quantification of MyoD and MyoG present in original Fig. 2d-f (Fig. 3c-e in revised manuscript) was performed on fresh isolated EDL myofibers, but not cultured myofibers. To our knowledge, very rare frequency of MyoD⁺ and MyoG⁺ cells occurs in fresh myofibers isolated from adult resting muscles. The

EdU incorporate assay has been reproduced and the new results were included in the revised manuscript (Fig. 4 and Supplementary Fig. 6). As we have submitted very clear images, we were also wondering if the PDF conversion has reduced the quality of the images.

3. Supp Fig. 2a the wild type controls need to be shown along with the Pten ko sections.

Authors' Response: We have added the image of MyoD and MyoG staining for WT control in the revised manuscript (Supplementary Fig. 5b).

4. The authors have identified a potentially important regulatory pathway to maintain quiescent satellite cells. However, the data presented do not sufficiently distinguish between a requirement of Pten for cell survival as opposed to a requirement for Pten to maintain quiescence. Additional data will support the authors' conclusions and provide an important contribution to the field.

Authors' Response: We appreciate the reviewer's comment. As detailed in earlier comments, we have now provided more convincing data to exclude the possibility of cells death (by TUNEL) and confirm differentiation and fusion (EdU plus dystrophin labeling) (Fig. 3,4 and Supplementary Fig. 4,5). These data conclusively demonstrate a critical requirement of Pten in maintaining quiescence of muscle stem cells. We appreciate the reviewer's comment that "the finding provides an important contribution to the stem cell field".

References

1. Kim D, Hwang I, Muller F, Paik J. Functional regulation of FoxO1 in neural stem cell differentiation. *Cell Death Differ.* **22**, 2034-2045 (2015).
2. Jeon JH, Suh HN, Kim MO, Ryu JM, Han HJ. Glucosamine-induced OGT activation mediates glucose production through cleaved Notch1 and FoxO1, which coordinately contributed to the regulation of maintenance of self-renewal in mouse embryonic stem cells. *Stem cells and development* **23**, 2067-2079 (2014).
3. Kitamura T, *et al.* A Foxo/Notch pathway controls myogenic differentiation and fiber type specification. *J. Clin. Invest.* **117**, 2477 (2007).
4. von Maltzahn J, Jones AE, Parks RJ, Rudnicki MA. Pax7 is critical for the normal function of satellite cells in adult skeletal muscle. *Proc. Natl. Acad. Sci. U. S. A.* **110**, 16474-16479 (2013).
5. Relaix F, Zammit PS. Satellite cells are essential for skeletal muscle regeneration: the cell on the edge returns centre stage. *Development* **139**, 2845-2856 (2012).
6. Mitchell KJ, *et al.* Identification and characterization of a non-satellite cell muscle resident progenitor during postnatal development. *Nat. Cell Biol.* **12**, 257-266 (2010).
7. Sampaolesi M, *et al.* Cell therapy of α -sarcoglycan null dystrophic mice through intra-arterial delivery of mesoangioblasts. *Science* **301**, 487-492 (2003).
8. Murphy MM, Lawson JA, Mathew SJ, Hutcheson DA, Kardon G. Satellite cells, connective tissue fibroblasts and their interactions are crucial for muscle regeneration. *Development* **138**, 3625-3637 (2011).
9. McCarthy JJ, *et al.* Effective fiber hypertrophy in satellite cell-depleted skeletal muscle. *Development* **138**, 3657-3666 (2011).
10. Buckingham M, Relaix F. The role of Pax genes in the development of tissues and organs: Pax3 and Pax7 regulate muscle progenitor cell functions. *Annu. Rev. Cell Dev. Biol.* **23**, 645-673 (2007).

11. Basch ML, Bronner-Fraser M, García-Castro MI. Specification of the neural crest occurs during gastrulation and requires Pax7. *Nature* **441**, 218-222 (2006).
12. Wen Y, Bi P, Liu W, Asakura A, Keller C, Kuang S. Constitutive Notch activation upregulates Pax7 and promotes the self-renewal of skeletal muscle satellite cells. *Mol. Cell. Biol.* **32**, 2300-2311 (2012).
13. Keefe AC, *et al.* Muscle stem cells contribute to myofibres in sedentary adult mice. *Nat. Commun.* **6**, (2015).
14. Pawlikowski B, Pulliam C, Dalla Betta N, Kardon G, Olwin BB. Pervasive satellite cell contribution to uninjured adult muscle fibers. *Skeletal muscle* **5**, 1 (2015).

REVIEWERS' COMMENTS:

Reviewer #1 (Remarks to the Author):

In the revised manuscript, the authors have provided new data showing the direct interaction between FoxO1 and RBPJk in primary myoblasts, and the interaction was significantly enhanced by either activation of Notch signaling or overexpression of a constitutively active FoxO1-ADA.

Reviewer #2 (Remarks to the Author):

I would like to commend the authors for the significant efforts they put in addressing my concerns. I am now satisfied that this paper is worthy of publication.

Reviewer #3 (Remarks to the Author):

The authors have addressed the concerns of the reviewer and strengthened the data supporting their conclusions.